# Scalable physical source-to-field inference with hypernetworks

**Berian James**\*  
*Technical University of Denmark & Pioneer Centre for AI*

*berian@tem.energy*

**Stefan Pollok**\*  
*Technical University of Denmark*

*stefanpollok94@gmail.com*

**Ignacio Peis**  
*Technical University of Denmark & Pioneer Centre for AI*

*ipeaz@dtu.dk*

**Elizabeth Louise Baker**  
*Technical University of Denmark & Pioneer Centre for AI*

*eloba@dtu.dk*

**Jes Frellsen**  
*Technical University of Denmark & Pioneer Centre for AI*

*jefr@dtu.dk*

**Rasmus Bjørk**  
*Technical University of Denmark*

*rabj@dtu.dk*

**Reviewed on OpenReview:** *https://openreview.net/forum?id=EvfwGpo135*

## Abstract

We present a generative model that amortises computation for the field and potential around e.g. gravitational or electromagnetic sources. Exact numerical calculation has either computational complexity $\mathcal{O}(M \times N)$ in the number of sources $M$ and evaluation points $N$, or requires a fixed evaluation grid to exploit fast Fourier transforms. Using an architecture where a hypernetwork produces an implicit representation of the field or potential around a source collection, our model instead performs as $\mathcal{O}(M + N)$, achieves relative error of $\sim 4\% - 6\%$, and allows evaluation at arbitrary locations for arbitrary numbers of sources, greatly increasing the speed of e.g. physics simulations. We compare with existing models and develop two-dimensional examples, including cases where sources overlap or have more complex geometries, to demonstrate its application.

## 1 Introduction

Physics-informed machine learning has proven to be a useful tool for modelling functions. In this work, we apply it to the problem of learning fields. This poses extra methodological challenges, since not only do we need to incorporate physical constraints, but we also require that we can compose the output functions linearly corresponding to the superposition of fields. Fields are the critical observable in most disciplines of physics. The gravitational field, magnetic and electric fields, as well as various quantum mechanical fields, have provided fundamental advances in the description of the physical world. Evaluating fields numerically is also crucial to the development of real-world technologies, e.g. in renewable or fusion energy systems. However, current simulation approaches are bounded by their scaling when faced with a high number of field sources (i.e. non-pointlike objects) or fine-grained field resolution, for which seminal numerical methods—the Fast Multipole Method (FMM) and Fast Fourier Transform (FFT) algorithms (Greengard & Rokhlin, 1987;

---

\*Equal contribution

| Property | Models | | Baseline models | | |
| --- | --- | --- | --- | --- | --- |
| | Fourier | FC+ILR | FC+INR | Linear | Exact |
| $\mathcal{O}(M + N)$ scaling | ✓ | ✓ | ✓ | ✓ | × |
| Principle of superposition | ✓ | ✓ | × | ✓ | ✓ |
| Admits physical interpretation | ✓ | × | × | × | ✓ |

Table 1: Comparison of desiderata across different explored models within the template architecture. The specification of models and baseline models is given in Section 5.2: FC+ILR is a fully-connected network with a hypernetwork-trained final layer (an 'implicit linear representation'); FC+INR is a standard fully-connected implicit neural representation hypernetwork.

Engheta et al., 1992)—either impose restrictive assumptions or fail to efficiently handle arbitrary source configurations.

We propose to tackle *source-to-field* inference—i.e. generating the field at any point around or inside an extended physical source, given its properties—using statistical learning, and in doing so, we uncover a set of distinct challenges that require innovative solutions. First, the model must exhibit exceptional flexibility, being capable of learning a function space that accurately represents fields generated by an arbitrary number of sources with varying properties. This demands a framework that goes beyond traditional grid-based learning approaches and that adapts to arbitrary source configurations. Second, we must rigorously maintain the *principle of superposition*. In this context, this means ensuring that each source contributes to the field in a manner that is both independent and linear. This requirement is not just a mathematical convenience but a physical necessity, reflecting the fundamental linear nature of how fields interact and combine. Lastly, the model must be informed by established principles of physics, for example Maxwell's equations, which for magnetic fields guarantee a divergence-free field.

To address these challenges, we propose a general architecture encompassing two models and two baseline variants (cf. Table 1), designed to:

- generate continuous field functions at arbitrary locations using hypernetworks (Ha et al., 2017);

- achieve linear scaling with the number of sources and evaluation points by designing hypernetworks whose outputs are combined linearly; and

- incorporate relevant physical properties, such as enforcing conservative fields by modelling the field as the gradient of its scalar potential.

By satisfying these criteria, our models significantly alleviate the computational cost of high-fidelity field simulations and open new directions for physical modelling through statistical learning. This approach advances the intersection of machine learning and physics, enhancing the applicability of generative models to real-world physical systems.

## 2 Background

### 2.1 Source-to-Field Inference

Our focus is on classical physical fields: the magnetic field generated by any electrical current or source with a magnetisation vector $\mathbf{M}$, or similarly an electrical field generated by electric charges; and the gravitational field generated by any object with a mass. In this work, we take as an example the magnetic field, but the results generalise directly to other fields.

It can be shown that the magnetic field $\mathbf{H}$ at a position $\mathbf{r}$ generated by an object with a magnetisation at position $\mathbf{r}'$ is given by (Jackson, 1999)

$$\mathbf{H}(\mathbf{r}|\mathbf{M}, \mathbf{V}') = -\frac{1}{4\pi} \int \underbrace{\left( \frac{\mathbf{r} - \mathbf{r}'}{\|\mathbf{r} - \mathbf{r}'\|^3} \nabla \cdot \right)}_{\equiv \mathbb{D}(\mathbf{r} - \mathbf{r}')} \mathbf{M}(\mathbf{r}') \, dV', \tag{1}$$

where $\|\cdot\|$ denotes the Euclidean norm and the term in parentheses acts like an operator $\mathbb{D}$ taking the divergence ($\nabla\cdot$) of the source magnetisation $\mathbf{M}$, and where the integral is evaluated over the source (with volume $V'$). The mathematical action of the operator is to take the magnetisation vector at each location $\mathbf{r}'$ within the source and yield a rotated and scaled version of it at $\mathbf{r}$. The above equation is a consequence of satisfying Maxwell's equations for the magnetic flux density, Gauss's law for magnetism $\nabla \cdot \mathbf{B} = 0$ and Ampère–Maxwell law for magnetism $\nabla \times \mathbf{B} = \mu_0 (\nabla \times \mathbf{M})$ in the absence of currents and the definition of the magnetic field $\mathbf{H} = \frac{1}{\mu_0} \mathbf{B} - \mathbf{M}$.

In three dimensions, $\mathbb{D}$ is a real, symmetric $3 \times 3$ tensor with components (Smith et al., 2010a, Eqs. A5–6)

$$D_{ij}(\mathbf{r} - \mathbf{r}') = \frac{\delta_{ij}}{\|\mathbf{r} - \mathbf{r}'\|^3} - \frac{3(r - r')_i (r - r')_j}{\|\mathbf{r} - \mathbf{r}'\|^5}, \tag{2}$$

where the indices $i, j$ run pairwise over the Cartesian coordinates $x, y, z$ and $\delta_{ij}$ is the Kronecker delta, which is equal to 1 if the indices are equal and 0 otherwise. The matrix $\mathbb{D}$ is a purely *geometric* object. The total field at $\mathbf{r}$ is then the superimposed contribution integrated over the spatial extent of the source.

For a *uniformly* magnetised source, like a permanent magnet or a magnetic domain, the magnetisation $\mathbf{M}_0$ is constant across the object, and so the magnetic field vector at (a single) $\mathbf{r}$ is given by

$$\mathbf{H}(\mathbf{r}|\mathbf{M}, \mathbf{V}') = -\left( \frac{1}{4\pi} \int \mathbb{D}(\mathbf{r} - \mathbf{r}') dV' \right) \cdot \mathbf{M}_0 = -\mathbb{N}(\mathbf{r}, \mathbf{V}') \cdot \mathbf{M}_0, \tag{3}$$

where by linearity $\mathbb{N}$ remains a real, symmetric $3 \times 3$ object known as the *demagnetisation tensor*, the mathematical operator ($\cdot$) is matrix multiplication and $\mathbf{V}'$ is the geometry of the source (i.e. location $\mathbf{r}'$, volume, and shape). There exist closed-form expressions for the tensor $\mathbb{N}$ for ellipsoids (Joseph & Schlömann, 1964), prisms (Aharoni, 1998), tetrahedra (Nielsen et al., 2019), and cylinders (Nielsen & Bjørk, 2020; Slanovc et al., 2022) among others. The same principles apply for the gravitational field, with the divergence of the magnetisation replaced by the density of the object.

Since most objects consist of uniformly magnetised domains, for calculations it is advantageous to consider the magnetic field at a point $\mathbf{r}_n$ as being generated by a collection of $M$ sources that obeys the principle of superposition, contributing independently and linearly to the field at all locations,

$$\mathbf{H}(\mathbf{r}_n|\mathbf{M}, \mathbf{V}') = -\sum_{m=1}^{M} \mathbb{N}(\mathbf{r}_n, \mathbf{V}'_m) \cdot \mathbf{M}_m \quad \text{with } \{\mathbf{r}_n\}_{n=1}^{N}, \tag{4}$$

where $m$ is indexing the individual magnetic sources, the index $n$ specifies the evaluation point, and $\mathbf{M}_m$ indicates the magnetisation vector of the $m$-th source. If the field must be computed at $N$ points, the overall cost scales as $\mathcal{O}(M \times N)$, since each of the $N$ evaluations requires summing over the $M$ sources with a distinct demagnetisation tensor termed $\mathbb{N}_{mn}$. This becomes particularly problematic in dynamical systems, where sources evolve over time, and the field must be recomputed repeatedly. While the demagnetisation tensor $\mathbb{N}$ only has to be computed once if the geometry remains constant, the calculation of this magnetic field is numerically the most time consuming task in simulations of e.g. the time evolution of a collection of magnetic spins (Abert et al., 2013), and various computational frameworks have been developed for evaluating the aggregate field from many sources at many points by geometric decomposition (Bjørk et al., 2021; Ortner & Coliado Bandeira, 2020; Liang et al., 2023).

An additional property for fields can be exploited to ease training. Because the gravitational field, as well as the magnetic field in the absence of currents, is conservative, it can be derived from scalar-valued functions

$\phi$ called *potentials*, where $\mathbf{H} = -\nabla\phi$. It should also be noted that because the magnetic flux density is divergence-free, the scalar potential satisfies Laplace's equation, i.e.

$$0 = \nabla \cdot \mathbf{H} = \nabla \cdot (-\nabla\phi) = -\nabla^2\phi; \tag{5}$$

everywhere except the boundary of the source, where $\nabla^2\phi = -\nabla \cdot \mathbf{M}$ (Griffiths, 2013, Eq. 6.23).

## 2.2 Hypernetworks

The computational cost of evaluating Eq. 4 motivates the use of flexible machine-learning surrogates that can represent the field function directly, allowing field values to be queried anywhere without recomputing each demagnetisation tensor.

Hypernetworks provide precisely such a mechanism. Originally introduced as neural networks that generate the weights of another network (Ha et al., 2017), they enable model parameters to be predicted dynamically rather than stored as fixed quantities. This allows us to model entire functions—such as a magnetic field—conditioned on physical context like magnetisation. Hypernetworks have powered advances in few-shot learning (Bertinetto et al., 2016), meta-learning (Munkhdalai & Yu, 2017), generative modelling (Dupont et al., 2022; Koyuncu et al., 2023; Peis et al., 2025), neural architecture search (Brock et al., 2018), and continuous scene representations, as in HyperNeRF (Park et al., 2021), where weights are conditioned on scene deformations.

Recent works have also demonstrated the potential of hypernetworks for modelling physical systems, where they can serve as fast surrogates for time-evolving or spatially distributed processes (Wan et al., 2023). In this work, we leverage hypernetworks to generate continuous field representations conditioned on a set of physical sources, enabling scalable and flexible source-to-field inference. Unlike traditional applications focused on discrete outputs or network weights, our approach produces entire continuous functions, making it well-suited to physical domains where evaluation at arbitrary spatial locations is required.

## 3 Method

The computational bottlenecks in classical field evaluation motivate the use of a learned surrogate model that can simultaneously i) encode an arbitrary number of sources into an efficient representation; and ii) query the field at any $\mathbf{r}_n$ for parameterised shapes, and ultimately from a learned embedding of sources. To address this, we propose a hypernetwork-based surrogate that maps source properties to continuous field functions, allowing evaluation at arbitrary locations and aggregation across sources. In the following, we consider static configurations of sources, i.e. configurations where the locations of the source points do not change in time, but where the magnetisation of the source points can change in time.

For magnetic fields, the functions $\mathbb{N}$ and $\mathbf{H}$ are continuous and differentiable, and so are likely candidates for approximation by a network trained on data examples generated from the analytically known forms for $\mathbf{H}(\mathbf{r}_n|\mathbf{M}, \mathbf{V}')$.

The challenge, therefore, lies in breaking the quadratic coupling between sources and evaluation points: instead of explicitly computing the demagnetisation tensor at every point $n$ for every source $m$, as in Eq. 4, we seek approximations for $\mathbf{H}(\mathbf{r}|\mathbf{M}, \mathbf{V}')$ that condition on a permutation-invariant aggregation of arbitrary sources, such that the field can be inferred in $\mathcal{O}(M + N)$. To this end, we propose the approximation

$$\mathbf{H}(\mathbf{r}_n|\mathbf{M}, \mathbf{V}') \approx f\left(\mathbf{r}_n \,\middle|\, \sum_{m=1}^{M} g(\mathbf{M}_m, \mathbf{V}'_m)\right) = \mathbf{a}(\mathbf{M}, \mathbf{V}') \cdot \boldsymbol{\psi}(\mathbf{r}_n) \quad \text{with } \{\mathbf{r}_n\}_{n=1}^{N}, \tag{6}$$

where $\boldsymbol{\psi}$ and $g$ (cf. Fig 1) are functions to be specified, and the basis weights $\mathbf{a}(\mathbf{M}, \mathbf{V}') = \sum_{m=1}^{M} g(\mathbf{M}_m, \mathbf{V}'_m)$ have to be evaluated only once for a source configuration as they are independent of $\mathbf{r}_n$. In our architecture, $g$ is implemented as a hypernetwork that outputs the weights or parameters for a field representation model $f$, which evaluates the field at query locations. We interpret $g$ as a representation for the sources, and it should be constructed so that a single representation of the source *collection*, i.e. the sum of the individual

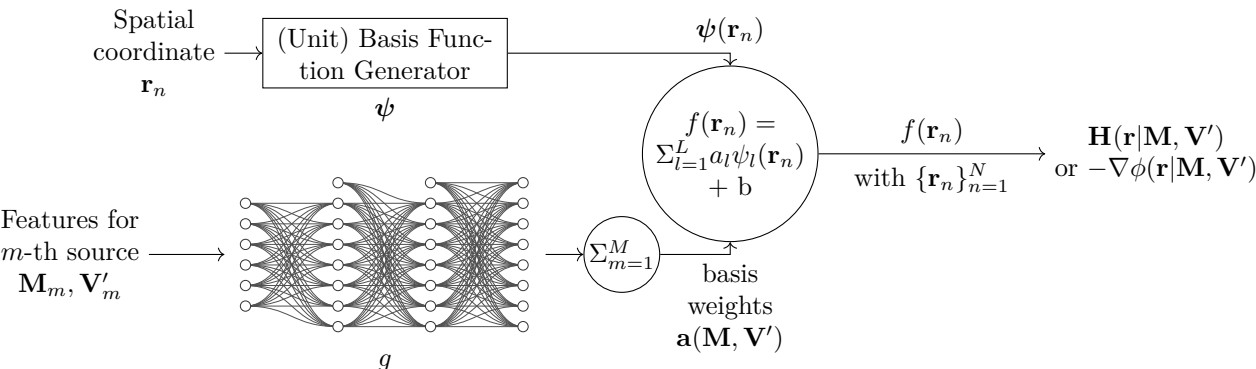

Figure 1: In the template architecture, a basis function generator $\boldsymbol{\psi} : \mathbb{R}^d \to \mathbb{R}^L$ expands the $d$-dimensional spatial coordinate $\mathbf{r}_n \in \mathbb{R}^d$ in a (fixed or learnable) basis to specified order $L$. The coefficients $\mathbf{a}_{1:L} \in \mathbb{R}^L$ of the expansion, with $a_0 = b$ representing the bias term, are learned as a hypernetwork $g : \mathbb{R}^v \to \mathbb{R}^{L+1}$ of the $v$-dimensional features of the source geometry $\mathbf{V}'_m$ and magnetisation $\mathbf{M}_m$, which is additive across the sources, giving the cumulative magnetic field or scalar potential at $\mathbf{r}_n$. It follows that once the basis weights $\mathbf{a}$ are accumulated for all $M$ sources, only $\boldsymbol{\psi}(\mathbf{r}_n)$ has to be recalculated in $f$ for any other evaluation point.

source embeddings, can accurately condition the approximating model $f$ for the field. These arbitrary collections of magnetic sources, with each source input with features like source position, volume, shape, and magnetisation, generate a single additive fixed-length vector from an arbitrary number of sources, rather than applying the network many times in parallel to the features of each input source. Especially note that $\mathbf{r}_n$ becomes independent of the summation index in Eq. 6.

Given access to a set of source configurations and their corresponding field observations, we train our model by minimising the Huber loss via stochastic gradient descent

$$\mathcal{L}_{\mathbf{H}} \; = \; \mathcal{L}_\delta \left[ \mathbf{a}(\mathbf{M}, \mathbf{V}') \cdot \boldsymbol{\psi}(\mathbf{r}_n) \; - \; \mathbf{H}(\mathbf{r}_n) \right] \quad \text{with} \;\; \delta = 1, \tag{7}$$

where the loss is averaged over all observation locations $\{\mathbf{r}_n\}_{n=1}^N$ and across mini-batches of source configurations. The Huber loss is used to handle the large dynamic range of field values arising from distance-dependent physical interactions. Compared to the mean squared error, it reduces the influence of large near-field errors while preserving sensitivity to small residuals, leading to more stable training across near-field and far-field regimes.

In the following sections, we explore different instantiations of this general framework. We begin by introducing a formulation that leverages the conservative nature of certain physical fields, enabling indirect field evaluation via scalar potentials. Next, we present additive hypernetworks, which enforce the principle of superposition by requiring $f$ to be linear in the aggregated source representation $g$. We then propose a physically motivated variant of $f$ based on Fourier expansions, before turning to fully connected alternatives that relax these inductive biases to achieve greater expressivity.

### 3.1 Direct and indirect evaluation of the field

As mentioned previously, the physics is equally well described by the magnetic scalar potential $\phi$. This observation allows us to model the field indirectly: instead of parameterising $\mathbf{H}$ itself, we can parameterise a scalar potential $\hat{\phi}$ with underlying functions $\hat{\mathbf{a}}$ and $\hat{\boldsymbol{\psi}}$ and recover the field via automatic differentiation

$$f\left( \mathbf{r}_n \,\bigg|\, \sum_{m=1}^M g(\mathbf{M}_m, \mathbf{V}'_m) \right) \; = \; -\nabla \hat{\phi}\left( \mathbf{r}_n \,\bigg|\, \sum_{m=1}^M g(\mathbf{M}_m, \mathbf{V}'_m) \right) = -\hat{\mathbf{a}}(\mathbf{M}, \mathbf{V}') \cdot \nabla \hat{\boldsymbol{\psi}}(\mathbf{r}_n). \tag{8}$$

This ensures that the resulting field approximation is strictly conservative by construction, and may present a superior target for learning. Additionally, since our training data are generated using MagTense (Bjørk et al.,

2021), which provides both the true field and its associated potential, we can incorporate this additional supervision into the loss function when the field is indirectly derived via the potential

$$\mathcal{L} \;=\; \gamma_{\mathbf{H}} \cdot \mathcal{L}_{\mathbf{H}} + \gamma_\phi \cdot \mathcal{L}_\phi, \tag{9}$$

where $\mathcal{L}_{\mathbf{H}}$ and $\mathcal{L}_\phi = \mathcal{L}_\delta(\hat{\phi} - \phi)$ are Huber losses with $\delta = 1$ applied to the field and potential predictions, respectively, and $\gamma_{\mathbf{H}}, \gamma_\phi \in \mathbb{R}_+$ are scaling weights that balance their contribution.

Additional physical constraints could influence our choices for the loss function. As mentioned previously, the magnetic scalar potential satisfies Laplace's equation, which could be incorporated via the loss (e.g., as in Pollok et al., 2023). However, in this work, we employ a loss function solely based on the known field / potential values in the provided data.

## 3.2 Additive hypernetworks for field inference

There are reasons to suppose that a representation like this might be possible. As mentioned previously, the physical sources obey the principle of superposition, contributing independently and linearly to the field at all locations. In the language of equivariance, this is to say linear additivity implies the sources will be permutation-invariant[1]. We provide additional discussion of this structure in Appendix B.2.

To guarantee the principle of superposition, then, we must require that $f$ be linear in $\sum_{m=1}^{M} g(\mathbf{M}_m, \mathbf{V}'_m)$, which is a significant constraint on possible architectures. This constraint ensures that the contribution of each source remains independent and additive in the latent space, faithfully mirroring the underlying physical laws. Geometrically, this implies that the space of source collections is a vector space, and that any $g_m$ is a coordinate in the space, weighting an abstract basis function representation $\boldsymbol{\psi}$ of the field around source collections. The architecture set out in Fig. 1 implements this construction.

We implement $g$ as a fully-connected hypernetwork that outputs the parameters of the basis functions to represent $f$. We leave open the specific parameterisation of $f$, and consider both a physically motivated choice and a more expressive, data-driven alternative to compare their performance.

## 3.3 Fourier Hypernetworks

An additive hypernetwork architecture guarantees the principle of superposition. In our initial setup, we focus further on the physical desirability of (orthogonal) decomposition of the potential, choosing a fixed set of basis functions $\boldsymbol{\psi}_p(\mathbf{k}, r_n) = [\cos(k_p r_n), \; \sin(k_p r_n)]^T$ in 1D or $\boldsymbol{\psi}_{pq}(\mathbf{k}, \mathbf{r}_n) = [\cos(k_p r_{nx})\cos(k_q r_{ny}), \; \sin(k_p r_{nx})\cos(k_q r_{ny}), \; \cos(k_p r_{nx})\sin(k_q r_{ny}), \; \sin(k_p r_{nx})\sin(k_q r_{ny})]^T$ in 2D, corresponding to a Fourier expansion of the scalar potential $\hat{\phi}$,

$$\hat{\phi}_{1\mathrm{D}}(\mathbf{r}_n) = \sum_{p=0}^{P-1} \mathbf{A}_p \cdot \boldsymbol{\psi}_p(\mathbf{k}, \mathbf{r}_n), \tag{10}$$

$$\hat{\phi}_{2\mathrm{D}}(\mathbf{r}_n) = \sum_{p=0}^{P-1}\sum_{q=0}^{P-1} \mathbf{A}_{pq} \cdot \boldsymbol{\psi}_{pq}(\mathbf{k}, \mathbf{r}_n). \tag{11}$$

In this model, $\boldsymbol{\psi}$ is fixed and corresponds to the Fourier basis evaluated at query locations, while $\sum_{m=1}^{M} g(\mathbf{M}_m, \mathbf{V}'_m)$ produces the corresponding coefficients $\mathbf{A}_p = [A_p, B_p]$, $\mathbf{A} \in \mathbb{R}^{P \times 2}$ in 1D and accordingly $\mathbf{A}_{pq} = [A_{pq}, B_{pq}, C_{pq}, D_{pq}]^T$, $\mathbf{A} \in \mathbb{R}^{P^2 \times 4}$ in 2D for wavenumbers $k_0 = 0$ and $\mathbf{k}_{1:P-1} \in \mathbb{R}_+^{P-1}$ up to order $(P-1)$ that modulate this basis. As $k_0$ already represents a constant component, we omit the bias term in $f$ for this model, and $g$ outputs here the basis weights $\mathbf{a} \in \mathbb{R}^L$, where $L = 2P$ in 1D and $L = 4P^2$ in 2D to represent $\mathbf{A}_p$ and $\mathbf{A}_{pq}$, respectively.

To understand why this reduces the complexity, note that the terms in $\boldsymbol{\psi}$ depend only on where the potential is to be evaluated, while all dependence on the configuration of the sources generating the potential, i.e. their

---

[1]Indeed Eq. 6 follows the form of a DeepSets architecture (Zaheer et al., 2017), however, we use $f$ and $g$ for the functions because $\phi$ is needed for the scalar potential.

geometry and magnetisation, is enclosed in the Fourier coefficients $\mathbf{A}$. As potentials are additive, and as the evaluation points stay constant in static problems, the Fourier coefficients $\mathbf{A}$ must be sums of the Fourier coefficients of each source. This means that if we compute these for every source, an operation of order $M$, and after adding them, we then only need to use these coefficients to evaluate the potential in the desired points, an operation of order $N$.

Rather than computing these coefficients via a Fourier integral, we amortise them through the hypernetwork, conditioned on the source configuration. We provide concrete examples of this construction in both 1D and 2D in Sec. 5.

Note that, while this is a non-linear function in the field location $\mathbf{r}_n$, it is linear in the weights as required to make the source collections a vector space. Unlike in a discrete Fourier transform, the wavenumbers $\mathbf{k}$ need not be tied to the resolution or domain window where the field is evaluated, and there is no *requirement* that they form an integer sequence—they could e.g. be set as learnable parameters—however, choosing integrally spaced wavenumbers makes the basis functions orthogonal, which can help eliminate degeneracies in the training landscape. There is a geometric correspondence to the use of random Fourier features in fully-connected networks (Rahimi & Recht, 2007); the use of $(P-1)$ random Fourier features projects the input coordinates into a high-dimensional random superspace spanned by $d_\psi \leq P-1$ independent basis functions, while an explicit choice of $(P-1)$ integrally-spaced wavenumbers projects to an orthogonal superspace of dimension $d_\psi = P-1$. Nevertheless, we found from initial testing that our FOURIER model performs better when $\mathbf{k}$ is set to a logarithmically-spaced sequence, where the upper and lower bounds are set empirically. Practical tuning considerations for this model are provided in Appendix C.2.

### 3.4  Networks without inductive biases

As an alternative, we relax the architectural constraints and allow the basis itself to be learned implicitly through a fully-connected network. In this setting, $g$ no longer outputs coefficients for a fixed basis, but instead generates parameters that directly shape the representation capacity of $\hat{\phi}$. To preserve the principle of superposition, we constrain $\sum g_m$ to produce only the weights $\mathbf{a}$ and bias $b$ to the output of $\boldsymbol{\psi}$. In this formulation, the activations of the final layer, of dimension $L$, serve as learned basis functions $\boldsymbol{\psi}(\mathbf{r}_n)$, and the potential can be expressed as

$$\hat{\phi}(\mathbf{r}_n) = \sum_{l=1}^{L} a_l \, \psi_l(\mathbf{r}_n) + b. \tag{12}$$

We refer to this construction as an *implicit linear representation* (FC+ILR), which preserves superposition while gaining expressivity through learnable basis functions. If we further relax this constraint, $\sum g_m$ may generate the full parameter set of the network, resulting in a highly expressive model akin to *implicit neural representations* (INRs; Sitzmann et al., 2020), where a hypernetwork (Ha et al., 2017) outputs all weights and biases.

We compare three instantiations of the general hypernetwork architecture, each defined by a different choice of $f$: FOURIER, Eq. 11, where $f$ is constructed from sine–cosine basis functions and the hypernetwork outputs the corresponding Fourier coefficients; FC+ILR, Eq. 12, where $f$ is a fully-connected network and the hypernetwork modulates only its final linear layer; and FC+INR, where the hypernetwork generates the full parameter set of $f$, albeit at the cost of losing superposition. Table 1 summarises the properties of these models, and in the experiments, we investigate their behaviour and performance.

## 4  Related Work

**Fast field evaluation and surrogate modelling.**  A key contribution of our work is the scaling behaviour of our generative model, which operates with a computational complexity of $\mathcal{O}(M+N)$, compared to the standard $\mathcal{O}(M \times N)$ in exact numerical simulation of fields such as the MagTense framework (Bjørk et al., 2021), or FFT-based methods requiring fixed evaluation grids. Unlike these methods, our model adopts amortised evaluation, significantly improving computational efficiency with only a small sacrifice in accuracy. This scaling perspective is also explored in the linearly constrained neural networks by Hendriks et al. (2020)

for the problem of magnetic fields. Our work focuses on learning representations for physical sources, which can be used in a generative model, and this presents a different set of challenges and opportunities.

**Physics-informed machine learning and basis representations.** Our work differs from general physics-informed neural networks (PINNs; Raissi et al., 2019; Karniadakis et al., 2021), which embed PDE constraints into the loss but typically require retraining for each new source configuration. While related work, e.g. Schaffer et al. (2023), has explored neural surrogates for demagnetising fields, our method generalises across source geometries without retraining and enables continuous-space evaluation. We further depart from standard PINNs by explicitly embedding the principle of superposition for a collection of sources and optimising the linear coefficients of fixed (FOURIER) / learned (FC+ILR) basis functions. This makes our approach closer in spirit to classical Galerkin methods, while remaining end-to-end trainable.

**Neural operators.** Similar in spirit to this work is that of neural operators (Kovachki et al., 2023). Neural operators work by learning an operator from an input function to an output function. For example, the Fourier neural operator framework (Li et al., 2021) uses a Fourier transform in the network architecture, effectively learning a truncated Fourier representation of a global convolution kernel corresponding to an integral operator, before inverse Fourier transforming to physical space. We instead keep the logic of the basis elements and coefficients separate, outputting the coefficients of the function directly. Doing so means we can get an additive structure simply by adding together learned coefficients. Moreover, since the hypernetwork architecture is not built on a specific transform, it allows for a choice in basis functions, or for the basis to be learned (see Fig. 1). Our setup is more in line with DeepONets (Lu et al., 2021), which is another popular variant of neural operators that weights high-dimensional embeddings of the query location with coefficients given by the embedded input function. However, instead of relying on fixed sensor locations for the input function, we obtain our coefficients from the features of single magnetic sources that fully describe the specific input function. This parametric description of the input makes our approach invariant to discretisation. Further, as we follow the principle of superposition, we can sum the coefficients of multiple *single-source* input function embeddings to directly infer the coefficients of *multi-source* input function embeddings. Hence, our training simplifies to learning from single-source samples while being capable of predicting multi-source samples with similar zero-shot performance.

**Conservative fields and architectural inductive biases.** Recent efforts to enforce physical laws through network design include divergence-free networks and conservation-driven approaches (Richter-Powell et al., 2022; Müller, 2023). These methods use differential forms or specialised architectural components to enforce symmetries. Our approach takes a complementary route: by constructing the magnetic field as the negative gradient of the scalar potential, we ensure a conservative field by design, which holds true in the absence of free currents, as in the present case of permanent magnets. Ghosh et al. (2023) similarly leverage harmonic representations in quantum systems; however, we focus on classical fields and emphasise linear additivity of sources rather than equivariant or gauge-theoretic formulations.

**Hypernetworks and physical models.** Our contribution extends the idea of hypernetworks (Ha et al., 2017), which have been increasingly applied in physical modelling (Pfaff et al., 2020; Cho et al., 2023; de Avila Belbute-Peres et al., 2021). We extend this direction by showing how additive hypernetworks can be designed to enforce superposition and enable scalable source-to-field inference. Unlike prior work using hypernetworks to model dynamical systems or PDE evolution, we target static field representations and develop a novel parametrisation (FOURIER, FC+ILR) that yields physically interpretable outputs with linear source aggregation.

## 5 Experiments

We implement experiments using finite magnetic sources, with fields and potentials derived from exact forms, either through the computational framework MagTense (Bjørk et al., 2021) or via direct implementation. Full details of the analytical expressions for spherical sources are given in Appendix B.1.

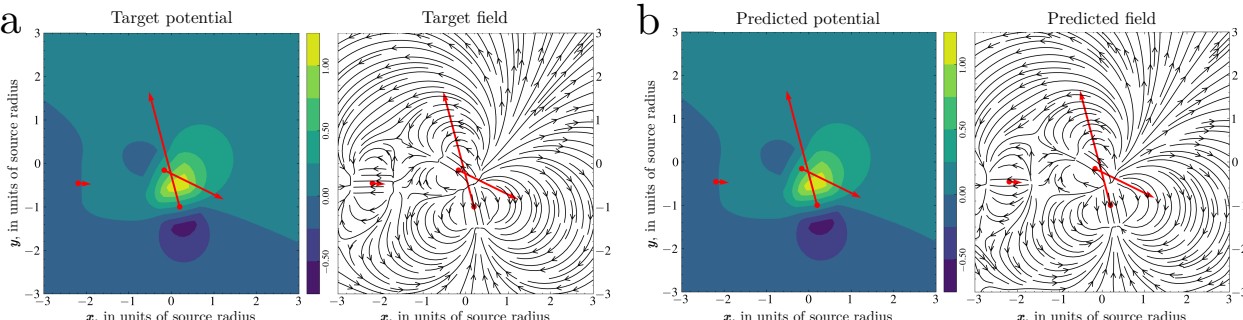

Figure 2: Magnetic potential and field for three finite circular sources (**a**) and their approximation by a small fully-connected network (**b**). Sources with locations $\mathbf{r}'$ and direction and magnitude of the magnetisation $\mathbf{M}$ shown by the red arrows, are randomly positioned within a $[-3, 3] \times [-3, 3]$ domain, in units of the source radius, with the potential and field generated on a regular $100^2$ grid.

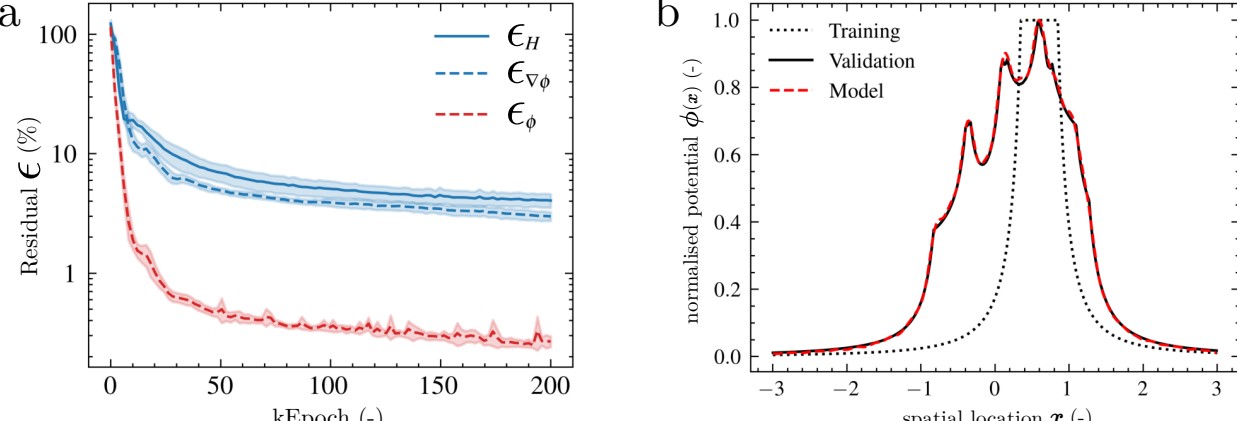

Figure 3: (**a**) Validation curves for predicting magnetic potential (red) and field (blue), either directly (solid) or indirectly via the potential (dashed), shown as relative error percentages. (**b**) Output of a 1D FOURIER hypernetwork trained on single-source potentials (dotted black), successfully generalising to the combined potential of six randomly placed sources.

We express the performance of the model prediction ($\hat{\phi}$ or $\hat{\mathbf{H}}$) as a relative error

$$\epsilon_\phi := \text{median}_{n \in N} \left| \frac{\hat{\phi}(\mathbf{r}_n) - \phi(\mathbf{r}_n)}{\phi(\mathbf{r}_n)} \right|; \quad \epsilon_{\mathbf{H}} := \text{median}_{n \in N} \frac{\|\hat{\mathbf{H}}(\mathbf{r}_n) - \mathbf{H}(\mathbf{r}_n)\|}{\|\mathbf{H}(\mathbf{r}_n)\|}. \tag{13}$$

We report the median instead of the mean because relative errors diverge as the true field value approaches zero; the median offers a more robust and representative summary of model performance across the domain.

Our code is available at `https://github.com/cmt-dtu-energy/hypermagnetics`.

## 5.1 Direct and indirect evaluation of the field

We first evaluate how well the architecture can represent the field around a single collection of sources, without requiring the model to learn source-conditioned representations—i.e., we represent $\psi$ as a fully-connected network and $\mathbf{a}$ as a learnable final linear layer to constitute $f$, similar to FC+ILR while omitting $g$ and the input from source features. When training models to amortise numerical computation, one can either learn to predict the field directly or predict the scalar potential and compute the field via numerical differentiation. It is therefore worth comparing both approaches. To demonstrate the ability of $f$ to represent the potential and field, we use a single sample consisting of three sources as shown in Fig. 2. Here, the 2D sources are represented as disks of equal radius $d_m = 0.5$ and are randomly placed within a $[-3, 3] \times [-3, 3]$ domain (in source-radius units). The input to $\psi$ are the 2D evaluation points $\mathbf{r}_n = (r_{nx}, r_{ny})^T$. The field

and potential of the source configuration are evaluated on a regular $100^2$ grid, with $N = 100^2$ random points reserved for validation. Two ensembles of fully-connected networks (width 32, depth 3) are trained with $\mathcal{L}_\mathbf{H}$ and $\mathcal{L}_\phi$ respectively for $10^5$ epochs using the Adam optimiser (Kingma & Ba, 2014) with a learning rate of $10^{-5}$. Each ensemble consists of 5 runs of different parameter initialisations, whose outputs are then averaged. The mean and variance of the ensemble validation errors during training are shown in Fig. 3a.

A model directly inferring the scalar potential ($\epsilon_\phi$) performs with 0.31% error. The same model can be used to infer the field by taking the numerical gradient of the potential ($\epsilon_{\nabla\phi}$), defined in the same way as $\epsilon_H$ but notated differently to express that the field evaluation is indirect; this reaches an error of 3.70%, and notably its validation curve follows the same shape during training. A separate model with the same hidden layer size that directly outputs the magnetic field from the fully-connected network ($\epsilon_H$) performs comparably to the indirect field method, reaching 4.70%, though with an apparently slower convergence. This clearly demonstrates generalisation to arbitrary field locations—but not to arbitrary source collections, as the source configuration was fixed here.

It is unsurprising that, for a fixed network size, field prediction is less accurate than potential prediction due to the instability of numerical differentiation. Moreover, training scalar outputs is typically easier than training vector outputs. Given that direct field prediction does not offer accuracy advantages, we adopt the potential-based strategy in the remainder of our experiments, which implies a simpler network architecture, results in conservative fields, and simultaneously enables direct evaluation of the potential, which is itself of practical interest.

| Model | $M = 1$ | $M > 1$ |
|---|---|---|
| FOURIER | 4.74 ($\pm$ 0.38) | 5.73 ($\pm$ 0.45) |
| FC+ILR | 4.38 ($\pm$ 0.32) | 4.76 ($\pm$ 0.39) |
| FC+INR | 3.68 | – |
| LINEAR | 70.0 | 71.2 |

Table 2: Out-of-sample relative error $\epsilon_\phi$ (%) for predicting the potential generated by magnetised 2D disks of equal radius.

## 5.2 Evaluation of arbitrary source configurations

While the previous experiment demonstrated generalisation across spatial locations, achieving the desired $\mathcal{O}(M + N)$ scaling requires that the model also generalise across arbitrary source configurations. To evaluate this, we generate a dataset of up to $10^4$ source collections, each with random positions within a $[-3, 3] \times [-3, 3]$ domain and magnetisations sampled from a scaled normal distribution with scaling factor $c = 1/\pi$. As sources are 2D disks of fixed radius ($d_m = 1$), $\mathbf{V}'_m$ is fully defined with the location of the source $(r'_{mx}, r'_{my}, 0)^T$ and its magnetisation $\mathbf{M}_m = (M_{mx}, M_{my}, 0)^T$. Hence, the input to $g$ is four-dimensional $[M_{mx}, M_{my}, r'_{mx}, r'_{my}]^T$ ($v = 4$). To leverage the linearity of superposition, we train only on single-source examples, i.e. $M = 1$. Each training example is evaluated at $N = 32^2$ randomly sampled potential points across the domain. At inference time, multiple source embeddings are aggregated before computing the field. For visualisation purposes, we use a fixed $128^2$ grid in this section.

All models share a common hypernetwork $g$, implemented as a fully-connected network of depth 3 and width defined as a multiple of its output size, which is the number of parameters in the last layer of $\boldsymbol{\psi}$. The networks are trained with a stepwise learning rate schedule using Adam (Kingma & Ba, 2014) for 5,000 epochs per step at learning rates $10^{-\{3,4,5,6\}}$. To ensure fair comparison, model sizes are kept similar—generally around 20M parameters—so that the overall number of parameter-epochs remains approximately constant. Detailed configurations are summarised in Table 4 in the Appendix.

### 5.2.1 The FOURIER network

For the FOURIER model, the ability to generalise to an arbitrary number of sources follows directly from the linearity of the hypernetwork outputs in the inference network. This is illustrated in Fig. 3b, where a 1D version of the model with 32 modes is trained solely on single-source examples. At inference time, weights from multiple sources are aggregated *before* evaluating the potential, demonstrating the use of superposition.

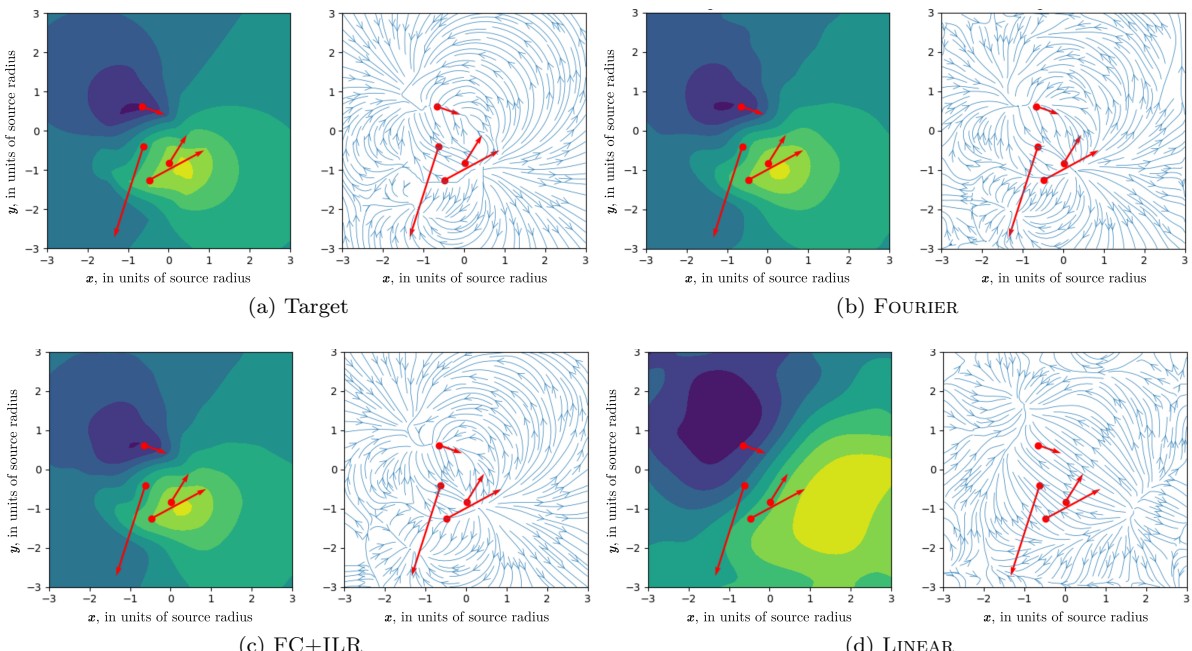

Figure 4: Multiple-source inference. (**a**) Ground-truth potential and field from four randomly placed circular magnets. (**b-d**) Predictions from models trained only on single-source examples, evaluated by first aggregating source representations before computing spatial values.

In two dimensions, the FOURIER model takes the form

$$\hat{\phi}(\mathbf{r}_n) = \sum_{p=0}^{P-1} \sum_{q=0}^{P-1} A_{pq} \cos(k_p r_{nx}) \cos(k_q r_{ny}) + B_{pq} \sin(k_p r_{nx}) \cos(k_q r_{ny})$$
$$+ C_{pq} \cos(k_p r_{nx}) \sin(k_q r_{ny}) + D_{pq} \sin(k_p r_{nx}) \sin(k_q r_{ny}), \tag{14}$$

where $k_p$ and $k_q$ denote the wavenumbers with $k_0 = 0$ and $k_p = 10^{-2.9+3.65*(p/(P-1))}$, while $P$ controls the expansion order. We set $k_q = k_p$ to ensure a square basis, so the total number of Fourier coefficients—and hence hypernetwork outputs—is $L \sim P^2$, comparable in size to the weights of a single linear layer of width $P$. Empirically, the expansion performs well with as few as 16 modes per dimension. For results in Table 2 and Fig. 4, we use $P = 32$ modes.

### 5.2.2 The FC+ILR model

For the 2D setting, Fig. 4 illustrates the ability of the FC+ILR model to infer the potential and field around finite multiple sources. We emphasise both that i) the model is trained exclusively on single-source examples, and that ii) at inference time, the source representations are computed individually and aggregated *before* evaluating the field at any location.

### 5.2.3 LINEAR and FC+INR baselines

The FC+INR model lacks architectural constraints enforcing superposition, and thus fails to generalise to multi-source configurations when trained only on single-source examples. Table 2 summarises out-of-sample relative errors $\epsilon_\phi$ across all models. While FC+INR performs best on single-source inputs (3.68% error), it cannot be evaluated on multi-source data.

As a further baseline, we implement a minimal LINEAR model, where both $\psi$ and $g$ are single linear layers (depth 1) without any non-linear activation function, scaled to match parameter count. The spatial coordinate and source features undergo only linear transformations before being concatenated in $f$. Despite

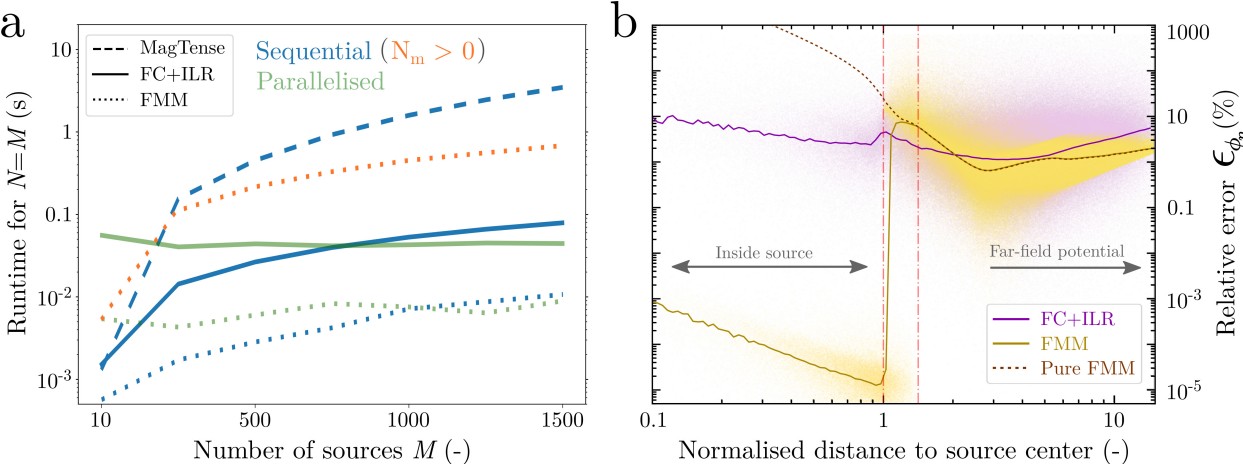

Figure 5: (**a**) Runtime as a function of the number of sources $M$ and keeping $N = M$, e.g. evaluation of $\phi$ in the centre of each source. The blue coloured lines show the sequential evaluation time to make the underlying computational complexity of each method apparent. The green lines show the runtimes when parallelising the workload on the GPU (NVIDIA GeForce RTX 5090) with FC+ILR and on 96 threads of the CPU (Intel® Xeon® Gold 6248R) in the case of FMM. The increased runtime visualised by the orange line originates from the additional correction required for evaluation points lying inside other sources in case sources overlap or $\phi$ is evaluated on a regular grid, which is denoted by $N_m > 0$. Notice that this only affects the evaluation time of FMM. (**b**) Relative error for single evaluation points $\epsilon_{\phi_n}$ from single-source configurations ($M = 1$) grouped by distance to the source centre normalised by the side length $s_x$ of the source. Here, the data consists of $10^3$ 2D potentials generated by single prism-shaped source configurations with varying side lengths ($s_x = s_y \in [0.12, 0.48]$) in a $[-1.2, 1.2] \times [-1.2, 1.2]$ domain. Each validation sample is evaluated at $32^2$ randomly sampled potential points across the domain. The FMM method used as a benchmark is shown in yellow, while we depict in brown the "pure" FMM without any corrections for the physical extension of the magnetic source. FC+ILR, depicted in purple, outperforms FMM close to the edges of the sources.

satisfying superposition and $\mathcal{O}(M + N)$ scaling, it lacks sufficient expressiveness, yielding relative median errors around 70%.

## 6 Experiments using a larger collection of sources and scaling behaviour

To test the model's applicability to more complex geometries, we extend our setup to prism-shaped sources, for which the magnetic scalar potential is known analytically (James et al., 2025). We use the FC+ILR model for this section, as it achieved the best overall trade-off in earlier experiments (see Table 2). The model is trained on $2 \times 10^5$ 2D potentials generated by single sources shaped as prisms with varying side lengths ($s_x = s_y \in [0.05, 0.5]$) with random positions in the $xy$-plane in a $[-1.25, 1.25] \times [-1.25, 1.25]$ domain. The 2D magnetisations are sampled from a scaled normal distribution with scaling factor $c = 10$. In contrast to Sec. 5.1, the input to $g$ is now five-dimensional $(M_{mx}, M_{my}, \mathbf{r}'_{mx}, \mathbf{r}'_{my}, s_{mx})^T$ to fully describe the source configuration ($v = 5$). Each training example is evaluated at $32^2$ randomly sampled potential points across the domain. Similar to Sec. 5.2, we train our model in this section with a stepwise learning rate schedule using Adam (Kingma & Ba, 2014) for $\{50, 100, 25, 25\}$ epochs at learning rates $10^{-\{2,3,4,5\}}$. We use the loss function defined in Eq. 9 with $\gamma_{\mathbf{H}} = 0.25$ and $\gamma_\phi = 1$. The hypernetwork $g$ is implemented as a fully-connected network of depth 3 and width 800, while $\boldsymbol{\psi}$ is a fully-connected network of depth 3 and width 400. Once the model is trained, it can be used in a versatile manner, as shown in the following experiments in Sec. 6.1 and in Sec. 6.2.

To benchmark our model, we compare against the Fast Multipole Method (FMM), in the open-source implementation by the Flatiron Institute (Cheng et al., 1999; Greengard et al., 2002; Greengard & Rokhlin,

1997; Greengard et al., 2002). The FMM method, which here is utilised in its 2D version, computes N-body interactions by compressing the interactions between clusters of source and target points at a hierarchy of scales. This means that the field can be computed as order $\mathcal{O}((M+N)\log(1/\tau))$ where $\tau$ is the desired relative precision, which we set to $10^{-5}$ for all experiments.

However, it remains necessary to correct the potential computed by FMM within each source, as the FMM method assumes point dipoles or collections of these, which within each source provides an incorrect potential as shown with the "Pure FMM" method in Fig. 5b. This correction cannot be applied in a manner that produces an overall correct potential, as the far-field dipole potential cannot be matched with an internally correct potential computed using the known analytical formulas for the potential generated by a rectangular prism. However, ignoring this, we apply the correction to FMM to get the correct potential within each source. This can be done in two ways: i) If the field is evaluated, and this is done in the centre of each source (i.e. $\mathbf{r}_n = \mathbf{r}'_m$), the field can easily be computed from the magnetometric demagnetization factor $\mathbb{N}_{mn}$, which for the field in 2D for symmetric objects is simply $1/2$ and the field is derived to $\mathbf{H}_n = -1/2\mathbf{M}_m$, and in 3D this factor is $1/3$ for symmetric objects. When evaluating the potential only in the centre, then FMM outputs the correct value ($\phi_n = 0$) for symmetric objects, and no correction is required. ii) However, when the field or potential is evaluated across a grid including points within each tile, this simple magnetometric correction is not applicable. We denote the number of evaluation points inside a source $m$ with $N_m$, which evaluates to $N_m > 0$ in this case. It follows that the field or potential must be calculated analytically using the known expressions (Smith et al., 2010b; James et al., 2025) within each point in the tile, which will add another $\sum_{m=1}^{M} N_m$ operations to the runtime of FMM as shown with the orange line of Fig. 5a.

When comparing the runtimes in Fig. 5a, we can see the expected quadratic complexity $\mathcal{O}(M \times N)$ for the classical evaluation method done using MagTense and a scaling of $\mathcal{O}(\lambda_m M + \lambda_n N)$ for the other methods, however with different scaling factors: $\lambda_{m,\text{FMM}} = \log(1/\tau) + \sum_M N_m$ and $\lambda_{n,\text{FMM}} = \log(1/\tau)$, and $\lambda_{m,\text{FC+ILR}} = L$ and $\lambda_{n,\text{FC+ILR}} = L + 1$. When parallelising the workload for FC+ILR, we obtain a flat runtime which becomes beneficial for around $M + N > 2*750$, as long as the problem fits on GPU memory. For larger-scale problems, we expect the runtime to follow the theoretical $\mathcal{O}(M+N)$ profile. Similarly, FMM exhibits a flat runtime for parallelisation across the CPU.

In Fig. 5b, we compare the performance of the method proposed here to FMM in configurations of prism-shaped magnetic sources. The relative errors of each evaluation point $\epsilon_{\phi_n}$ are grouped by the normalised distance to the source centre to highlight the peculiarities of each method. FC+ILR relative error rates are centred between $1\%$ and $10\%$ for the whole distance range. The best performance can be seen for distances between $1.5s_x$ and $6s_x$ from the source centre, while the relative error goes up to $6\%$ for the far-field potential. Interestingly, the error rates approach high error rates of $10\%$ when reaching the centre of the sources. This is arguably due to the potential approaching $0$ close to the source centre, hence small errors become relatively large. FMM, corrected by MagTense inside the sources, reaches relative error rates of around $10\%$ close to the edges of the magnetic source, when higher order poles come into play. For the far-field potential, when the potential from a dipole becomes a better approximation, $\epsilon_\phi$ grouped by the normalised distance stays below $2\%$. Nevertheless, the relative error becomes larger when increasing the normalised distance beyond $3$, which is because absolute error diminishes at a slower rate than the $\|\mathbf{r}\|^{-1}$ scaling of the 2D potential values.

Further, we consider two different scenarios for comparing the overall performance of our model to FMM, namely one with overlapping sources and one with quadtree structures. Additionally, a small-scale comparison to Fourier neural operators and how this framework can be applied to these experiments is provided in Appendix E.

## 6.1 Overlapping sources

It is of great interest to be able to model overlapping sources, as this is often the only way of modelling complex geometries without resorting to discretising the geometry to a large degree. Because fields and potentials are additive, a geometry with e.g. a hollow cylinder can be realised by starting with a cylinder and then on this overlapping a smaller cylinder with an opposite magnetisation compared to the larger cylinder. This will result in a cancellation where the cylinders overlap, and this produces the correct field

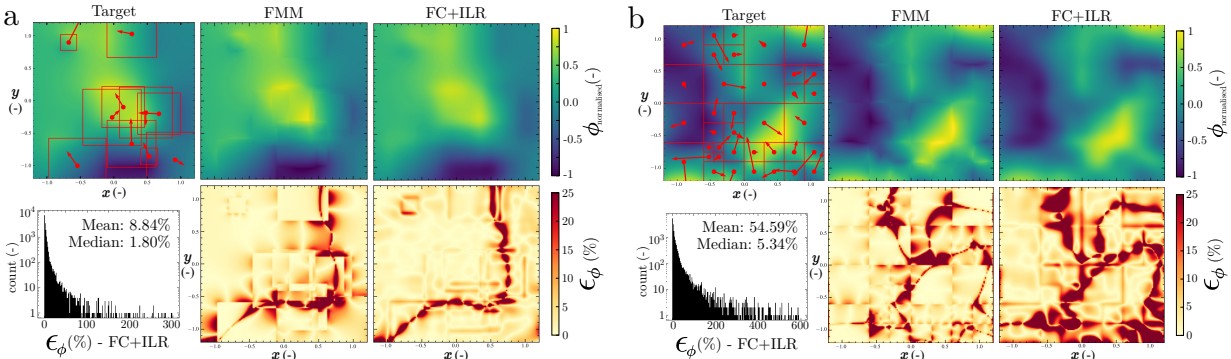

Figure 6: Scalar potential from sampled configurations of magnetic sources used in our experiments, comparing ground truth $\phi$ with the predictions from FMM, $\hat{\phi}_{\text{FMM}}$, and our FC+ILR model, $\hat{\phi}_{\text{FC+ILR}}$. The prism-shaped sources are depicted in red with the arrows indicating their magnetisation $\mathbf{M}$. At the bottom, a histogram of the relative error of $\hat{\phi}_{\text{FC+ILR}}$ is shown along with $\epsilon_\phi$ spatially resolved for $\hat{\phi}_{\text{FMM}}$ and $\hat{\phi}_{\text{FC+ILR}}$. The spatial extensions are normalised and hence dimensionless. (**a**) Normalised scalar potential of 10 overlapping sources. (**b**) Normalised scalar potential of a quadtree structure with 37 sources.

with only two sources. The alternative strategy of discretising the hollow cylinder with e.g. prisms or tetrahedra, which would have resulted in a much larger number of sources. Therefore, it is computationally desirable to be able to correctly model overlapping sources.

In this experiment, we calculate the magnetic scalar potential from a varying number of sources ($M = 10, 50, 250, 1000$) on a regular grid of $32^2$ evaluation points. The median relative error in the scalar potential $\epsilon_\phi$ (vs. the exact solution computed using MagTense) is presented in Table 3. An example with $M = 10$ sources is depicted in Fig. 6a. Note that here the evaluation points are a regular grid of $128^2$ values, which is chosen for visualisation purposes. Each entry in the table represents an average over 10 different test samples, with its standard deviation in brackets. Notably, the potential error remains below 5% for this scenario with FC+ILR, regardless of the number of sources—in line with the small-scale experiments with spherical sources in Table 2. The reason for the slightly larger error of the FMM method is exactly that the dipole approximation is not applicable in the proximity of a physical magnetic source, which means that error accumulates when multiple sources overlap, exemplified at the bottom of Fig. 6a.

## 6.2 Quadtree structures

It is also of interest to consider the sources arranged in a way such that they span a volume of space in a non-overlapping way, and with more sources located at volumes where a high resolution is desired. Such a structure could be a quadtree structure, which we term a structure where space is subdivided into increasingly smaller prisms by splitting each prism into four smaller prisms, with the same aspect ratio, as depicted in Fig. 6b for $M = 37$. Such a configuration is important in e.g. micromagnetic simulations where a simulation domain can be meshed using rectangular prisms, with the smaller prisms resolving finer features such as grain boundary layers (Bjørk & Insinga, 2023).

In this experiment, we calculate the magnetic potential similarly from a varying number of sources ($M = 10, 50$) and evaluate it on a regular grid of $32^2$ points. When increasing the number of sources further, i.e. $M \gg 50$, the side lengths of some prism-shaped sources $s_x, s_y \ll 0.1$, and the scenario cannot be evaluated in a fair comparison, as the range is outside what FC+ILR has been trained for. The results in Table 3 clearly show that FMM outperforms our model for this scenario with $\epsilon_{\phi,\text{FMM}} < 3\%$. All evaluation points lie within a source and, as shown in Fig. 5b, FMM greatly benefits from the applied correction in this area. However, as $N_m \gg 0$, the runtime of FMM is around one order of magnitude larger than FC+ILR.

Our trained model stays below 10% relative error. The increased error compared to the scenario of overlapping sources arises from the prevalence of small sources with $s_x < 0.2$ as can be seen in Fig. 6b. This is further emphasised in Fig. 8a in the Appendix, where the relative error of single-source configurations is

| Metric | Model | $M = 1$ | Overlapping sources / Sec. 6.1 | | | | Quadtree / Sec. 6.2 | |
|---|---|---|---|---|---|---|---|---|
| | | | 10 | 50 | 250 | 1000 | 10 | 50 |
| $\epsilon_\phi$ (%) | FMM | **1.14** (± 0.28) | **2.21** (± 0.78) | 4.00 (± 0.84) | 6.14 (± 2.12) | 5.32 (± 2.30) | **2.75** (± 0.49) | **2.69** (± 0.47) |
| | FC+ILR | 2.21 (± 1.48) | 2.43 (± 0.54) | **3.05** (± 0.88) | **4.40** (± 1.53) | **4.21** (± 1.51) | 6.40 (± 2.79) | 5.70 (± 1.14) |
| $\mathrm{MAE}_\phi$ | FMM | 2.89 (± 9.19) | 6.62 (± 11.12) | 8.76 (± 8.82) | 12.42 (± 11.72) | 12.91 (± 10.71) | **7.89** (± 10.74) | **9.96** (± 16.28) |
| $(\times 10^{-3})$ | FC+ILR | **2.86** (± 5.93) | **5.76** (± 7.48) | **6.20** (± 5.59) | **8.32** (± 6.68) | **10.00** (± 7.84) | 13.50 (± 15.63) | 17.28 (± 21.31) |

Table 3: Mean and standard deviation of the relative error $\epsilon_\phi$ and the mean absolute error ($\mathrm{MAE}_\phi = N^{-1}\phi_{\max}^{-1}\sum_{n=1}^{N}|\phi_n - \hat{\phi}_n|$) across 10 different test samples for an increasing number $M$ of prism-shaped sources inside one sample. $\phi_{\max}$ is the absolute maximum value of the potentials in the respective experiment. The best performing method is marked in bold. The experiment with $M = 1$ refers to the single-source configurations used in Fig. 5b.

plotted using the FC+ILR model. The reason for the large error of the method proposed here is that it is hard for the neural network to learn the potential far from a source, where the potential is close to zero. The dynamic range of potential values for small prisms is simply too large for the model to learn them properly, which means that the predictions for sources with small side lengths become incorrect. As can be seen from Fig. 8b, it is the smallest sources that result in a large error in the prediction over the whole domain.

# 7 Limitations & Discussion

We finish by delineating what this architecture should be capable of *in principle*, against what we have concretely demonstrated through these experiments. This work develops models that provide inference from arbitrary collections of sources, but our experiments demonstrate this only in a limited range of evaluation domains, source volumes, and magnetisations. For practical results in (e.g.) magnetostatics, it is necessary to develop models in the same architecture trained with the maximum expected parameter range in such applications, while overcoming the difficulties to learn a large dynamic range of magnetic field values as discussed in the Appendix D. As it is known physically that the 2D far field, i.e the field measured far away from its source, falls off with $\|\mathbf{r}\|^{-2}$ and the 2D far-field potential follows a $\|\mathbf{r}\|^{-1}$ scaling, this could be explicitly incorporated into the learning process in future work.

When replacing the magnetisation with the mass as input to $g$, the same architecture can perform inference for the field and potential around gravitational sources. Further, developing these models as amortising replacements for infeasible numerical simulation requires extension to three dimensions. All models developed in this paper can be extended in this way; the FOURIER computation will then contain $(2L)^3$ terms, which makes the parameter requirements for the FC+ILR model seem more scalable, where only the input layers of $\psi$ and $g$ necessarily have to grow in size to incorporate additional geometric information and the magnetisation in 3D. From a learning perspective, for modelling 3D quantities properly, we also expect intermediate layer sizes and the output layer size $L$ to grow to have access to an increased number of basis functions. Moreover, we have not explicitly treated equivariance beyond noting a connection between the principle of superposition and the permutation invariance of sources.

As previously mentioned, this work develops models for static configurations. This architecture could be integrated iteratively into a time-varying problem, using the model output to update properties of the sources and then re-evaluating the field. However, it would be more natural to consider training neural operator(s) to evolve the system. As demonstrated, this model is most effectively used with collections of sources with spatial extent; for particle-like sources, FMM is a proven non-statistical algorithm with $O(M + N)$ scaling (Greengard & Rokhlin, 1987). Additionally, because the magnetic field depends on the geometry of the source, the network will have to be retrained for different source geometries, such as cylinders or tetrahedra. However, it is possible to envision that, as only the field near a source depends strongly on geometry, but further away all sources tend to the dipolar field Bjørk & d'Aquino (2023), a single network could be trained on different source geometries, but this is beyond the scope of this work.

Lastly, our particular choice of decomposition in trigonometric basis functions (Eq. 11), while physically motivated, may be improved by explicit weighting so that $\lim_{r \to \infty} \phi = 0$; more physical information known from Maxwell's equation, i.e. $\nabla \cdot (\mathbf{H} + \mathbf{M}) = 0$, may further improve training the proposed architecture; with defining $g(\mathbf{M}_m, \mathbf{V}'_m) = |\mathbf{M}_m| \cdot g(\mathbf{I}_m, \mathbf{V}'_m)$, where $\mathbf{I}_m$ is the unit vector pointing in the direction of $\mathbf{M}_m$, as we know from Eq. 3 that the field is linear in the strength of the magnetisation or directly applying rotations and translations in the vector space spanned by the abstract basis representation $\boldsymbol{\psi}$, may be another viable route to improve the proposed method. These limitations and ideas can be addressed with future work, which will be of interest to both the statistics and physics communities.

## 8 Conclusions

In this work, we addressed the problem of amortising field computation around physical sources by introducing a general hypernetwork-based architecture and three model variants, each balancing scalability, accuracy, and physical fidelity. Modelling via the scalar potential improves training efficiency and simplifies network design. Among the proposed approaches, the FOURIER and FC+ILR models achieve $\sim 5\%$ error while enabling field or potential inference for arbitrary source collections through the principle of superposition. While the physically motivated FOURIER model offers interpretability through its decomposition in trigonometric basis functions, it does not outperform the more flexible FC+ILR model and incurs greater architectural complexity and tuning overhead.

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

## A  Broader Impact Statement

Magnets are a critical energy and infrastructure technology because they convert between motion and electricity. The goal is to create permanent magnets that are as strong and as stable as possible: the stronger the field, the more efficient the conversion of energy; the more stable the magnet, the more versatile its applications across a range of temperatures and environments. Despite a sound analytical basis for atomic-scale physics, it is not possible to derive the magnetic properties of new materials or solve an inverse problem to design a material with a desired field. Our research could be used to provide much more rapid exploration of magnetic material and configuration possibilities, accelerating the development of these much-needed energy technologies.

The scaling improvements from our architecture enable compute savings for numerical work with data that satisfy the general physical principle of superposition. We hope that our research will enable highly accurate modelling with vastly less energy expenditure at inference time. Additionally, our research supports the goal of explainable inference for physical systems by pursuing amortising models grounded in physical theory. This should be seen as a desirable contingency against a purely empirical model, which may exhibit hidden performance biases resulting in poor or unpredictable performance for physical systems.

Nevertheless, there are risks from unintentional inaccuracy or imprecision in model output: at a minimum, the resource and opportunity cost of pursuing experimental and manufacturing effort in materials and configurations that cannot exhibit the performance the model claims. More seriously, successful model performance accelerating the development of energy technologies can lead to negative societal outcomes if those new technologies are not used responsibly or with appropriate safeguards. Certainly, this has been the case for energy technologies in the past, and in general, social and regulatory structures have emerged that can mitigate against such disasters.

## B  Theoretical Foundations and Extensions

### B.1  Details of Analytical Field Computation

For spherically-symmetric sources, the dipole field at $\mathbf{r}$ from a collection of $M$ point-like sources of radii $d_m$ with magnetic moments $\mathbf{m}_m = \frac{4}{3}\pi d_m^3 \mathbf{M}_m$ at positions $\mathbf{r}'_m$ is computed via the scalar potential (Jackson, 1999, p. 196)

$$\mu_0 \mathbf{H}_\odot(\mathbf{r}, \mathbf{M}, \mathbf{V}') = -\nabla \sum_{m=1}^{M} \underbrace{\overbrace{\frac{1}{4\pi|\mathbf{r}-\mathbf{r}'_m|^2}}^{\text{Surface of 3D ball}} \overbrace{\frac{\mathbf{m}_m \cdot (\mathbf{r}-\mathbf{r}'_m)}{|\mathbf{r}-\mathbf{r}'_m|}}^{\text{dipole term}}}_{\text{scalar potential } \phi(\mathbf{r},\mathbf{M}_m,\mathbf{V}'_m)} \text{ or } -\nabla \sum_{m=1}^{M} \frac{\mathbf{M}_m \cdot (\mathbf{r}-\mathbf{r}'_m)}{3}, \tag{15}$$

corresponding to outside and inside the source, respectively. The distinguishing feature of the dipole term is the $1/r$ dependence; higher multipole terms ($1/r^2$, $1/r^3$) will be present for specific shapes of sources, but

|  | FOURIER | FC+ILR (Sec. 5) | FC+ILR (Sec. 6) | FC+INR |
|---|---|---|---|---|
| Width of $\psi$ | 4×32×32 | 400 | 400 | 20 |
| Depth of $\psi$ | - | 3 | 3 | 3 |
| Width* of $g$ | 0.25 | 1.5 | 2 | 1.0 |
| Depth of $g$ | 3 | 3 | 3 | 2 |
| *Total Parameters* | *6.3M* | *1.5M* | *2.1M* | *1.7M* |
| Optimiser | | | Adam | |
| Average Learning Rate | 5E–4 | 1E–4 | 7.5E–4 | 1E–5 |
| $\gamma_{\mathbf{H}}$ | 1 | 1 | 0.25 | 1 |
| Steps (thousands) | 20 | 25 | 20 | 20 |
| GPUs | | | NVIDIA A100-SXM4-40GB ($\diamond$) | |
| Time (hours) | 12 | 6 | 40 | – |

Table 4: Collation of model parameters used for the experiments in Sec. 5 and Sec. 6 with the architecture developed in this work. If not specified otherwise, the same value is used for all experiments. A single entry in a row means the same value for all models. The Gaussian error linear unit (GELU) is used as the activation function in all fully-connected neural networks. The weights of the linear layers are randomly initialised from a uniform distribution over the interval $[\,-1/\sqrt{\text{fan\_in}},\ 1/\sqrt{\text{fan\_in}}\,]$, where 'fan\_in' describes the number of input features. (*) The (hidden layer) width of the hypernetwork $g$ is expressed as a multiple of its output size, which is the total number of learned parameters in the inference network $\psi$. ($\diamond$) For experiments in Sec. 6, we have instead used NVIDIA GeForce RTX 5090-32GB as the GPU.

the dipole term will dominate at larger $r$ (Bjørk & d'Aquino, 2023), although higher multipole terms are required near the source.

## B.2 On the principle of superposition

The principle of superposition implies that the field generated by a collection of sources can be obtained by summing the individual contributions of each source. In our architecture, this property is realised through a latent representation that is additive across sources. Specifically, each source embedding produced by the hypernetwork $g$ contributes linearly to the field representation $f$, such that the aggregation of embeddings corresponds to the aggregation of fields.

$$
\begin{array}{ccc}
\text{Set}\{\mathbf{H_r}(\mathbf{M}_m, \mathbf{V}'_m)\} & \xrightarrow{\ \Sigma_{m=1}^{M}\ } & \mathbf{H_r}(\mathbf{M}, \mathbf{V}') \\[2pt]
{\scriptstyle \mathbf{H_r}(\cdot)}\uparrow {\scriptstyle \forall m \in M} & & \uparrow {\scriptstyle \mathbf{H_r}} \\[2pt]
(\mathbf{M}, \mathbf{V}') = \text{Set}\{(\mathbf{M}_m, \mathbf{V}'_m)\} & \xrightarrow[\ \bigcup_{m=1}^{M}\ ]{} & \bigcup(\mathbf{M}_m, \mathbf{V}'_m)
\end{array}
$$

Figure 7: Commutative diagram for the construction of the field around a source collection via a fixed-length vector encoding. $\mathbf{H_r}$ describes here the evaluation of the magnetic field at point $\mathbf{r}$.

This structure is naturally expressed using the commutative diagram of Fig. 7, where the field produced by aggregating source embeddings is equivalent to aggregating the fields of individual sources.

## C   Experimental and Implementation Details

### C.1   Hyperparameters

We have implemented our models using JAX 0.4.25 (Bradbury et al., 2024) and Equinox (Kidger & Garcia, 2021), with training performed on the EuroHPC Karolina GPU cluster. Table 4 collates the model parameters used for the experiments in Sec. 5 and Sec. 6.

The output size of a hypernetwork is the number of parameters consumed by the main (inference) network, and this varies across the models as follows:

- The Fourier model is parameterised by the order $L$ of the expansion, and in two dimensions its inference 'network' requires $4L^2$ parameters;

- In the FC+ILR model, only the final layer of size $L$ of the inference network is set by the hypernetwork, so the total number of learnable parameters is the remaining layers of the inference network plus the hypernetwork size;

- In the FC+INR model, the total learnable parameters are those of the hypernetwork, which will quickly have a large output size as the inference network grows. However, because this model does not have the property of linear additivity that allows inference to generalise to multi-source examples, we have not tried to train a large network of this kind.

Consequently, we have found it easiest to express the hypernetwork width in units of its output size. The total number of parameters across both networks is then listed separately.

### C.2   Tuning tactics for the FOURIER model

To coax strong performance from the Fourier architecture, we use the following tactics, which run contrary to intuitions about Fourier series for sampled data:

**Selection of modes.** Our intuition for reconstructing a signal from sampled data would follow the constraints on wavelengths for the FFT—the shortest mode is set by the Nyquist sampling frequency, and the longest mode is set by the domain window size. In fact, the domain size is irrelevant here *because the underlying function is not periodic*, and the modes should be significantly longer. In practice, we let the upper and lower bounds for the modes be free parameters, and use a logarithmically-spaced sequence (of length the order of the expansion), affixing the zero-frequency mode to automatically include the Fourier bias terms.

**Irregularity of data sampling.** The periodic Fourier modes mean that training on a regular grid encourages overfitting: the reconstruction oscillates wildly even a short distance from the node points. Good practice would in any event suggest that training with data that are irregularly sampled over the domain would be more reliable, and this is the case.

**Joint fitting of potential and field.** Given the practical interest in the field resulting from the model output, we find improved performance using a loss with a term for the target potential and an equally-weighted term fit to the target field, evaluated via the gradient of the potential.

### C.3   Influence of the training objective

We encountered difficulties in learning the large dynamic range of the 2D potential in Sec. 6, which arises primarily from variations in source size. Additionally, the $1/\|\mathbf{r}\|$ scaling for 2D potentials complicates learning for our method. To guide practitioners, we have collected our approaches to cope with this challenge in the following:

**Modelling the log-potential.** It is tempting to train on the logarithm of the 2D potential. However, as the potential is defined in $\mathbb{R}$, we then need to differentiate between positive and negative values. Further, as our method relies on the superposition principle in the outputted scalar potential, any non-linear output normalisation scheme is prohibited. This also holds for other specialised normalisation schemes.

**Learning correction to dipolar field.** As we can fairly quickly (see blue dotted line Fig. 5a) derive the 2D potential from a point-like dipole, we tried to learn only the corrections to it for the prism-shaped sources in Sec. 6 to eliminate the $1/\|\mathbf{r}\|$ scaling of the far-field potential. However, preliminary experiments did not show any improvement in our metrics.

**Overpopulating data with large error.** We skewed our training data to comprise more data with small-sized magnetic sources. It had no major effect on the performance of our method.

**Larger strength of the magnetisation.** While comparing the different models with 2D disks of fixed radius in Sec. 5.2, magnetisation components $\mathbf{M}_x$ and $\mathbf{M}_y$ were drawn from a scaled normal distribution with scaling factor $c = 1/\pi$. As magnetic fields and scalar potential are relative quantities, it is possible to multiply the magnetisations by a constant and remove it afterwards by dividing model predictions by the same factor. To partly compensate for the smaller-sized magnets introduced in Sec. 6, we increase the scaling factor to $c = 10$, which leads to an improved performance of the trained model. This points towards the fact that our method for the currently chosen hyperparameters has a preferred field range.

## D  High error rates from small sources in FC+ILR

The source size has a substantial influence on the metric $\epsilon$ used in our experiments in Sec. 6. To analyse the root cause of this, we plot $\epsilon_\phi$ against the prism-shaped source side lengths $s_x, s_y$ in Fig. 8a. The data points correspond to the median relative error of $10^3$ 2D potentials generated by single prism-shaped source configurations ($M = 1$) with varying side lengths ($s_x = s_y \in [0.12, 0.48]$) in a $[-1.2, 1.2] \times [-1.2, 1.2]$ domain. Each validation sample is evaluated at $32^2$ randomly sampled potential points across the domain. It can be clearly seen that when $s_x < 0.2$, then $\epsilon_\phi > 10\%$, while we obtain consistently low relative error for larger source sizes. To further investigate where this behaviour originates, we plot the relative error for single evaluation points $\epsilon_{\phi_n}$ from single-source configurations ($M = 1$) grouped by distance to the source centre normalised by the side length $s_x$ of the source in Fig. 8b. Across the whole range of the normalised distance, the relative error $\epsilon_{\phi_n}$ of sources with $s_x < 0.2$ is almost an order of magnitude larger compared to the error of sources with $s_x > 0.4$.

When we look at four random samples with side lengths $s_x < 0.2$ from this dataset in Fig. 9, we spot two mechanisms responsible for large error rates: i) For small sources lying at the edge or corner of the sampling domain, the normalised distance to their source centre becomes large and the dynamic range becomes considerably larger than in other samples in the dataset, which is apparently difficult for the neural network to learn, and ii) the relative errors become generally large in the axis perpendicular to the magnetisation, where the potential approaches values close to 0. The latter effect is also apparent for samples with larger source sizes, but far less pronounced.

## E  Comparison with Fourier neural operator

In machine learning, the current state-of-the-art for approximating the solution of a family of parametric PDEs is neural operator learning (Kovachki et al., 2023), which maps a function of the PDE parameters at the input to the solution function at the output. An efficient and scalable variant of this approach, with strong empirical performance, is the Fourier neural operator (FNO) framework (Li et al., 2021), which we compare to our experiments in Sec. 6 in the following. In contrast to MagTense (Bjørk et al., 2021), FMM (Greengard & Rokhlin, 1987), and our method, where the input function is fully defined by source features $(\mathbf{M}, \mathbf{V}')$, FNO requires a spatial map of the magnetisation to learn a truncated Fourier representation of a global convolution kernel. To make FNO efficient, the input and output maps further have to be discretised on a

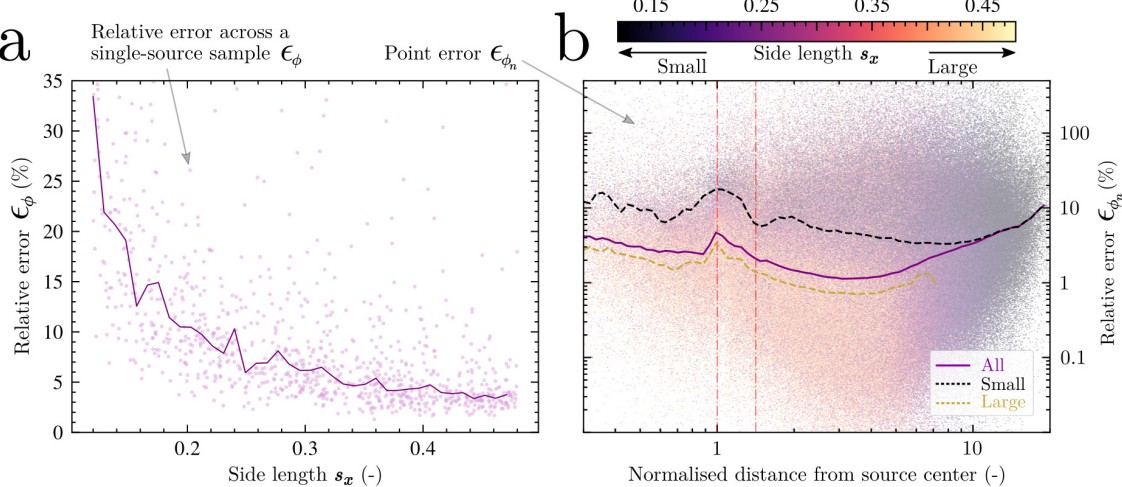

Figure 8: (**a**) Relative error $\epsilon_\phi$ as a function of prism-shaped source side lengths. The 2D potentials generated by these configurations ($M = 1$) are the same data as used in Fig. 5b. (**b**) Relative error for single evaluation points $\epsilon_{\phi_n}$ from single-source configurations ($M = 1$) grouped by distance to the source centre normalised by the side length $s_x$ of the source. The overall performance of FC+ILR is depicted in purple. The dotted lines show the grouped median relative error for small, i.e. $s_x < 0.2$, and large, i.e.@ $s_x > 0.4$, source sizes in black and yellow, respectively.

uniform, fixed grid so that FFT can be applied. As depicted in Fig. 10, a preprocessing step to represent the five-dimensional source features to a 2D spatial map of the magnetisation is straightforward. However, this representation faces challenges in precisely capturing the geometry of the magnetic source and high resolutions at the input are required.

We use the FNO implementation from NeuralOperator (Kossaifi et al., 2026), where we adjust the size of the FNO model to have 1.2 million total parameters, which is in the order of trainable parameters available for our method (see Table 4). This is achieved by setting the number of modes to keep in the Fourier layer to 16 in each dimension, the number of Fourier layers to 4, and the number of channels to 32. The default parameters are used otherwise. Following one of the examples in the documentation of NeuralOperator, we train FNO with a stepwise learning rate scheduler using AdamW (Loshchilov & Hutter, 2017) for 30 epochs per step at learning rates $5 \times 10^{-\{3,4,5,6\}}$. As the loss function, we use the mean squared error between predicted and ground-truth scalar potential values.

The dataset with $2 \times 10^5$ samples is taken from Sec. 6. However, we need to adjust the input and output data, which need to be 2D spatial maps of equidistant sample points, to enable FFT. Therefore, we represent the source features $(\mathbf{M}, \mathbf{V}')$ as a regular grid for each magnetisation component, as shown in Fig. 10. The magnetic scalar potential $\phi$ is then evaluated at a regular grid of resolution $32 \times 32$. As the input resolution can be scaled with respect to the output resolution, we train two versions of FNO, namely $\text{FNO}_{32}$ and $\text{FNO}_{128}$, where we choose the magnetisation map to be represented as a regular grid of resolution $32 \times 32$ and $128 \times 128$, respectively. For $\text{FNO}_{128}$, we have to change the resolution scaling factor from $[1, 1, 1, 1]$ to $[0.5, 0.5, 1, 1]$ inside the FNO framework to match the resolution of the potential at the output. By doing this, we want to investigate how much FNO depends on the input function resolution to capture the exact shape of the source geometry.

The performance of FNO compared to FMM and FC+ILR on selected scenarios is presented in Table 5. It can be seen that both FNO models perform slightly worse than the other two methods on the single-source test samples ($M = 1$), and that $\text{FNO}_{128}$ outperforms $\text{FNO}_{32}$ here. For multi-source samples ($M > 1$), FNO shows much higher errors compared to the other methods, which is not surprising, as the FNO models have only seen single-source samples during training and have no inherent mechanism to allow for zero-shot multi-source predictions.

| Metric | Model | $M = 1$ | Overlapping sources / Sec. 6.1 $M = 10$ | Quadtree / Sec. 6.2 $M = 50$ |
|---|---|---|---|---|
| $\epsilon_\phi$ (%) | FNO$_{32}$ | 9.53 ($\pm$ 6.99) | 46.7 ($\pm$ 21.4) | 66.5 ($\pm$ 14.1) |
| | FNO$_{128}$ | 4.44 ($\pm$ 2.26) | 45.5 ($\pm$ 10.4) | 63.8 ($\pm$ 20.3) |
| | FMM | **1.14** ($\pm$ 0.28) | **2.21** ($\pm$ 0.78) | **2.69** ($\pm$ 0.47) |
| | FC+ILR | 2.21 ($\pm$ 1.48) | 2.43 ($\pm$ 0.54) | 5.70 ($\pm$ 1.14) |
| MAE$_\phi$ ($\times 10^{-3}$) | FNO$_{32}$ | 6.25 ($\pm$ 9.99) | 82.3 ($\pm$ 67.6) | 165 ($\pm$ 128) |
| | FNO$_{128}$ | 4.08 ($\pm$ 7.84) | 88.9 ($\pm$ 85.0) | 159 ($\pm$ 131) |
| | FMM | 2.89 ($\pm$ 9.19) | 6.62 ($\pm$ 11.1) | **9.96** ($\pm$ 16.3) |
| | FC+ILR | **2.86** ($\pm$ 5.93) | **5.76** ($\pm$ 7.48) | 17.28 ($\pm$ 21.3) |

Table 5: Mean and standard deviation of the relative error $\epsilon_\phi$ and the mean absolute error (MAE$_\phi$ = $N^{-1}\phi_{\max}^{-1}\sum_{n=1}^{N}|\phi_n - \hat{\phi}_n|$) across 10 different test samples for selected scenarios of prism-shaped sources inside one sample. $\phi_{\max}$ is the absolute maximum value of the potentials in the respective experiment. The best performing method is marked in bold. The experiment with $M = 1$ refers to the single-source configurations used in Fig. 5b.

In Fig. 11, we have depicted a sample with $M = 10$ overlapping sources from the experiment described in Sec. 6.1 and a quadtree structure with $M = 50$ from Sec. 6.2, along with the predictions from FNO$_{128}$.

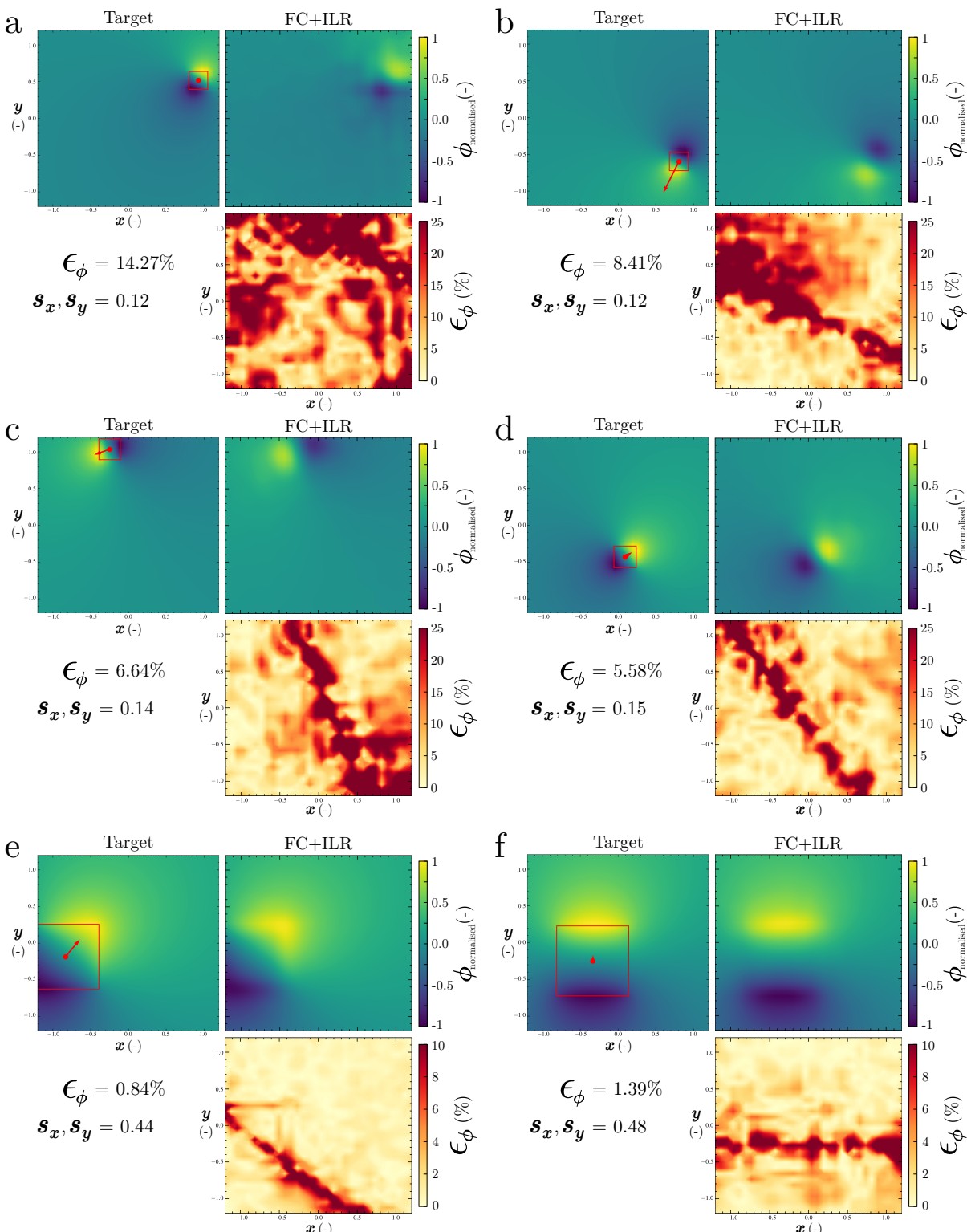

Figure 9: (**a-d**) Four samples of single prism-shaped source configurations ($M = 1$) with side lengths $s_x < 0.2$. (**e-f**) Two samples of single prism-shaped sources with a side length of $s_x > 0.4$. Along with the target and source configuration, the prediction from FC+ILR, the respective spatially-resolved relative error $\epsilon_\phi$ is presented on a regular grid with $32^2$ evaluation points. The spatial extensions are normalised and hence dimensionless.

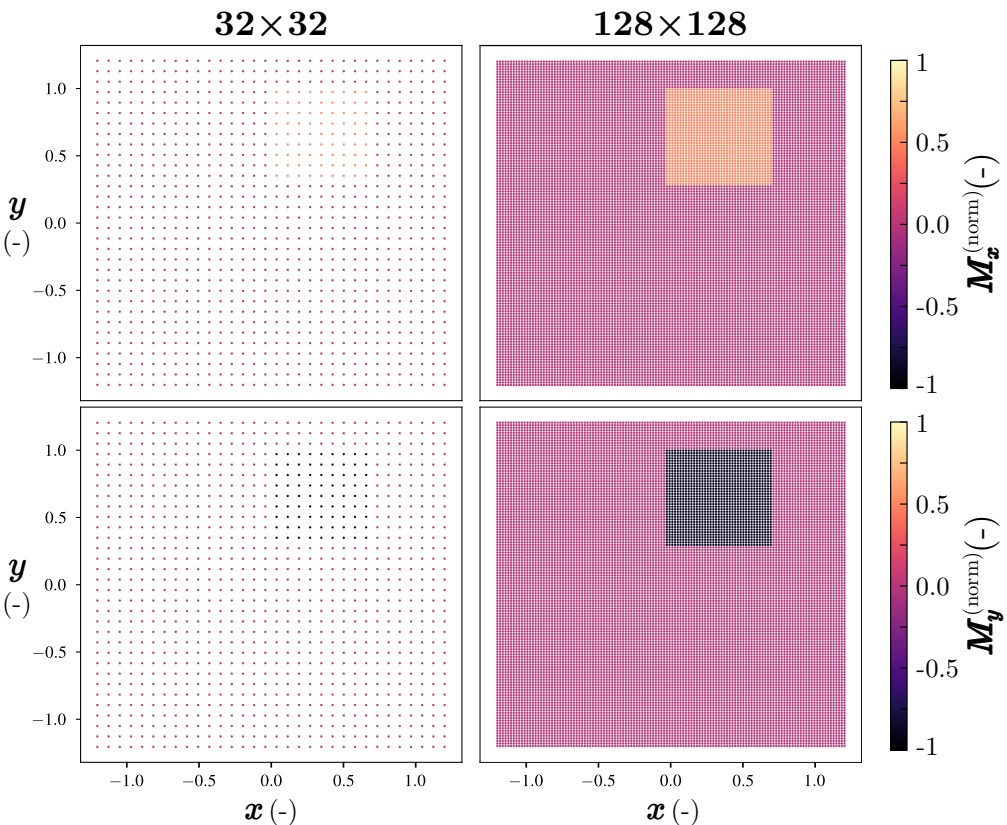

Figure 10: The input map of a single-source sample of the experiments in Sec. 6 with a five-dimensional input vector $(M_{mx}^{(\mathrm{norm})}, M_{my}^{(\mathrm{norm})}, r'_{mx}, r'_{my}, s_{mx})^T = (0.50, -0.87, 0.33, 0.64, 0.36)^T$, is preprocessed to the 2D spatial input map of the magnetisation required for FNO. On the left-hand side, we present a low-resolution map, i.e. $32 \times 32$ equidistant points, and a high-resolution map, i.e. $128 \times 128$, for the $x-$ and $y-$ component of the normalised magnetisation $M_{\square}^{(norm)} = M_{\square} \cdot \|\mathbf{M}\|_{\max}^{-1}$. $\|\mathbf{M}\|_{\max}$ is the maximum magnetisation strength in the respective experiment. Note that this normalisation is done for visualisation purposes only, while unnormalised magnetisation values are required for training. For the task of approximating the corresponding magnetic scalar potential at the output, it is essential to know precisely where the magnetic source is located, which becomes less prominent for lower resolutions.

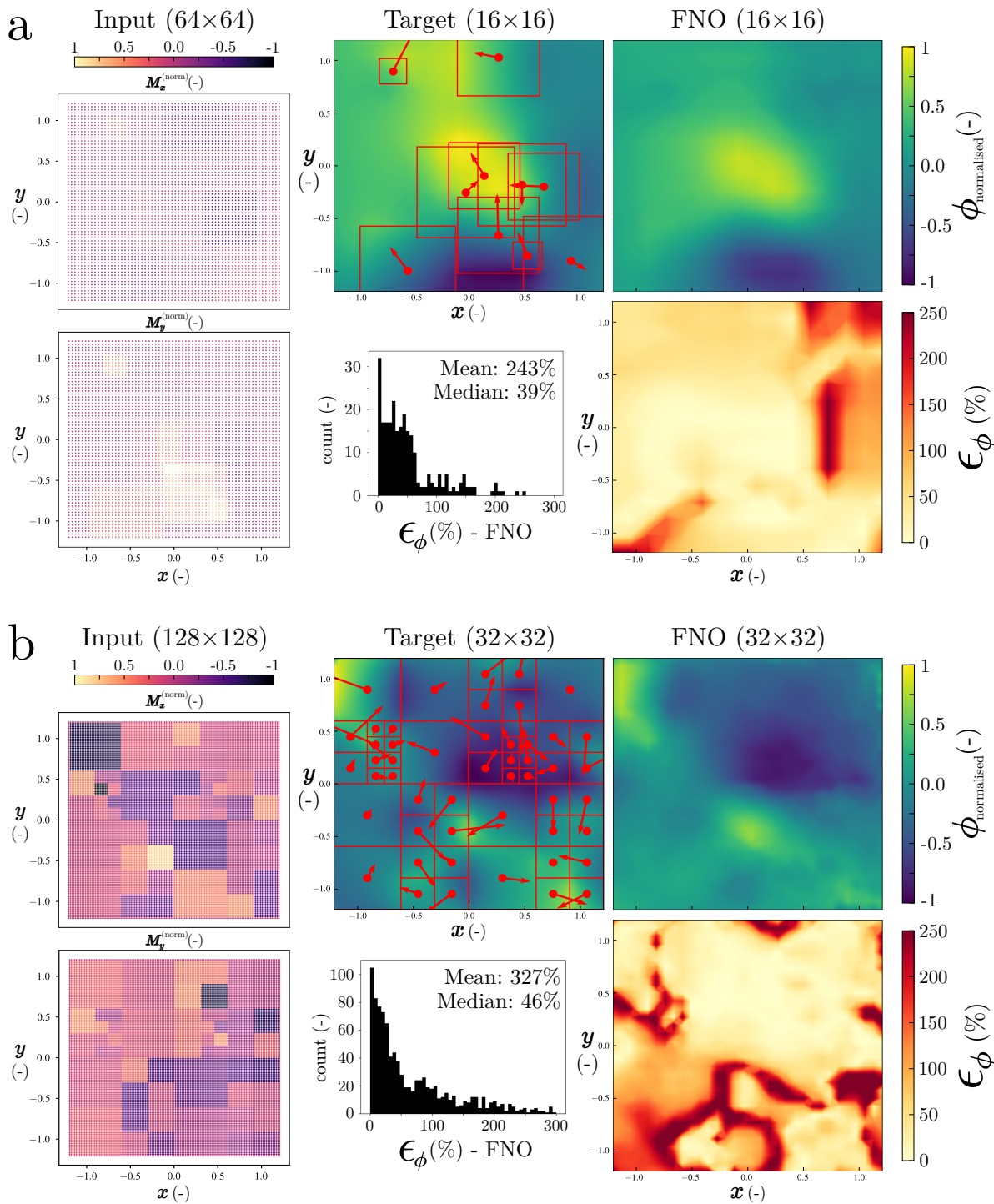

Figure 11: Scalar potential prediction from $FNO_{128}$ in zero-shot multi-source experiments. (**a**) Normalised scalar potential of 10 overlapping sources. As $FNO_{128}$ downsamples the resolution from the input to the output by a factor of 4, we can also infer a $16 \times 16$ scalar potential from two $64 \times 64$ magnetisation maps. (**b**) Normalised scalar potential of a quadtree structure with 50 sources. The 2D spatial maps of the two components of the magnetisation are shown on the left-hand side. The two remaining plots on top depict the ground-truth potential $\phi$ (*Target*), where the outline of the prism-shaped sources along with their magnetisations **M** are marked in red, next to the prediction $\hat{\phi}$ of $FNO_{128}$. At the bottom, a histogram of the relative error of $\hat{\phi}_{FNO}$ is shown along with $\epsilon_\phi$ spatially resolved for $\hat{\phi}_{FNO}$.

