# OpenReview forum: "Scalable physical source-to-field inference with hypernetworks"
_TMLR — Accepted by TMLR_

### Review · Reviewer_2oVn · 2025-11-21

**Summary Of Contributions:**

The authors propose a data-driven, machine-learning approach to approximating physical fields induced by the presence of one of more sources. While the primary focus is on magnetic fields, the approach is likely to generalize to other settings. The primary benefit of the method is in its scaling properties with respect to the number of sources, $M$, and evaluation points, $N$, compared to existing numerical algorithms. In general, experiments demonstrate that the method produces reasonably small error rates compared to existing approaches, including the Fast Multipole Method, while improving efficiency.

Strengths:

* The approach appears to perform fairly well in the settings considered and the authors investigate several methodological variations, which is informative.

* The incorporation of the principle of superposition into the network design is both interesting and an important contribution. In addition to its modeling advantages, it allows for elements such as hypernetwork inference parallelization.

* The theoretical ability to model various kind of source geometries and seamlessly accumulate disparate source impacts on the resulting field is important.

Weaknesses:

* In the paper, the claim that the method "incorporate[s] relevant physical properties, such as enforcing divergence-free fields through novel Fourier hypernetworks." does not appear to be supported. It is possible that I am missing something, but I do not see why any of the proposed approaches guarantee satisfaction of Gauss's law for magnetism. The use of a potential does ensure satisfaction of the Ampère–Maxwell law for the static setting, but this is also never clearly discussed.

* There are a number of areas where clarity could be improved.

* Several interesting experiments are performed, but there are a few important areas that are under-explored that would better support the claims of the paper. Two pertinent examples include whether the network can generalise across source geometries without retraining, as claimed in the paper, and the impact that items such as network size, basis dimension, and training steps, among others have on performance.

* The choice of median relative error as the primary evaluation metric is a bit odd. Perhaps this is the standard measure for this field. However, having several error perspectives, including something as simple as total or average non-relative error would be helpful.

**Additional Comments:**

Clarifications and Minor Comments

1) A bit of motivation for why the Huber loss is used over something like standard MSE would be useful.

2) When using a Fourier representation for the electric potential, could you use derivatives of the Fourier representation to derive the field, rather than relying on numerical differentiation?

3) There are typos "supoerposition", "bracktes", and "because his model" in the body and appendix.

4) Table 1 appears quite far from it's first refence on Page 6.

5) Other parameters, beyond learning rate, for the Adam optimizer are not specified anywhere.

6) In Section 5.1, it is stated that an "ensemble of fully-connected networks" is trained. However, it is never concretely stated how the networks are ensembled. That is, are they summed, uniformly averaged, or something else?

7) The activation functions for the dense networks are never stated anywhere.

8) Equation 13 has extra white space after the $\Sigma$s that should be fixed.

9) It would be helpful to know how the plots for the field and potential are created. That is, what kind of interpolation (or other approach) is used between the evaluation points?

10) Are the "10 runs" referred to in Section 6.1 ten separate training runs? It would be helpful to be specific.

11) In Appendix B.1, the notation $\mathbf{m}_m$ clashes with $\phi(\mathbf{r}, \mathbf{M}_m, \mathbf{V}'_m)$.

12) Table 4 does not do a good job specifying what "Width" and "Depth" refer to. I think they are for $f$? Moreover, "Hnet" is not used anywhere in the body. Is this $g$?

**Audience:**

Yes

**Audience Explanation:**

I think the approach proposed in this paper is both interesting and important. I am confident that this paper will be of interest to some TMLR readers. However, the presentation and experiments need work to better support the authors' claims.

**Broader Impact Concerns:**

I have no broader impact or ethical concerns with this work.

**Claims And Evidence:**

No

**Claims Explanation:**

To be more specific, I think the current experiments are a good start, but the clarity of the presentation needs to be improved and there are opportunities to strengthen the results to fully support the claims of the paper. See Requested Changes for more details.

**Requested Changes:**

1) There is no concrete discussion around how $\mathbf{M}_m$ and $\mathbf{V'}_m$ are represented as inputs to the hypernetwork $g$. This is also true for the spacial coordinates, $\mathbf{r}_n$, as input to $f$. While $\mathbf{M}_m$ and $\mathbf{r}_n$ may be straightforward, they should still be discussed. However, specifying the source geometry is not trivial.

2) This work exclusively considers static field representations, but this is not made explicit until Page 4, well after the derivation of several relevant equations.

3) I am not an expert on Maxwell's equations. However, in the static setting I believe that a magnetic field needs to satisfy both Gauss's law for magnetism and the Ampère–Maxwell law. The latter, however, is never discussed concretely, nor is it considered in the method construction. This needs to be more clearly presented.

4) It's possible I have missed it, but I don't see where we get divergence-free fields from the constructions suggested in this work, Fourier or otherwise.

5) Based on the presentation (for example in Equation 1), it appears some non-dimensionalization has been applied such that the permeability of free space is 1. If so, this should be mentioned in some capacity.

6) The notation $\vert \cdot \vert$ in Equation (1) and others is never defined for vectors. I think this is the Euclidean norm, but am not sure. This notation also potentially clashes with $\Vert \cdot \Vert$ in Equation (12) if that notation is also the standard Euclidean norm. Both should clearly state what is meant by the notation and be unified if they represent the same thing.

7) The notation $\delta_{ij}$ is never defined.

8) There is disonance between the form of the Fourier expansion of the scalar potential "in $d$-dimensions" in Equation (10) and that presented in (13). A better unification and discussion of these is important.

9) At the end of Section 3.3, the discussion of $\mathbf{k}_l$ is not very clear. For example, it mentions that "integrally spaced wavenumbers makes the basis functions orthogonal", which seems to suggest that the $\mathbf{k}_l$ are integers, but this is not the case in a standard Fourier basis. Nor is it the case in the experiments (see Section 5.2.1). Appendix C.2 is also quite vauge in terms of what these values end up being in practice. It also states "In practice, we let the upper and lower bounds for the modes be free parameters" Does this mean they are learned during the training process or something else?

10) Neural operators and their relationship to this work should be discussed, at least briefly, in the Related Work section.

11) The errors presented in Equation (12) are useful. However, it is unclear whether these are the standard way to measure error for these problems. If so, a reference would be helpful. If not, some justification for why these formulations are used here should be given. Regardless, it would also be useful to present other views of error, such as total or average non-relative error.

12) The training setup of Section 5.1 is not very clear. Is $g$ being randomly initialized and fixed while training $f$? Is the Fourier or FC+ILR approach being applied or is $f$ operating completely alone with no input from $g$. If the latter, why is $g$ ever mentioned? This should be more clear.

13) In the second paragraph of Section 5.1, the statement "This clearly demonstrates generalisation to arbitrary field locations—but not to arbitrary source collections." appears. However, this is only comparing training curves in Figure 3a. Do the authors mean to say that these are validation errors as training proceeds (i.e. validation curves)? If so, this needs to be clarified, as training curves typically represent training loss.

14) In Section 5.2.3, the presentation of the setup for the "Linear model" is unclear. Is this simply the FC + ILR setup, but where the networks for $f$ and $g$ are restricted to linear transformations, as opposed to deep networks with non-linear activations? If so, this should be stated explicitly.

15) For the prism-shaped source setup at the beginning of Section 6, if we're doing a 2D simulation, why is the height of the prism ($s_z$) relevant? If it has specific implications for the experiments, this should be stated, otherwise its inclusion is confusing.

16) The notation of $N_{\text{tile}}$ and $N_{\text{tile}}^{(m)}$ are introduced without explanation on Page 11. This makes interpreting and understanding the results in this section difficult.

17) The training setups for Overlapping Sources and Quadtrees are never described. That is, are the models trained only on single source setups, as in previous experiments? Are the remainder of training parameters also the same?

18) In Section 6.2, in discussing the failures associated with $M=250$, the authors state that the model "can not be evaluated for the scenarios with $M = 250$ as the side lengths of some prism-shaped sources $s_x,s_y < 0.1$, which is simply outside the range FC+ILR has been trained for." This leaves the question of whether the model could simply be trained on prisms in this size range to overcome this issue unresolved.

19) In addition to the above, in Section 6.2, the authors state that "the dynamic range of potential values for small prisms is simply too large for the model to learn them properly." This statement is not well supported by the experiments. That is, it may just be that the size of the prisms is outside of the training range rather than the model being unable to learn them at all. There are no experiments showing that the model couldn't learn to represent them. Would a larger model suffer the same fate, do you need finer Fourier modes, etc.?

20) There are no experiments showing generalization across source geometries (cylinders to prisms, for example) without retraining. Nor are there any experiments showing that one can mix different kinds of source geometries during, for example, training.

21) In Appendix C.2, the importance of training on irregularly sampled points is stated, but in the experiments, it appears that all training data was sampled on a uniform grid. If this is not the case it should be made clear. Otherwise, the statements of this appendix need to be amended.

---

> ### Author Response · Authors · 2025-12-14
> **Response to Reviewer 2oVn (1/4)**
>
> Dear Reviewer 2oVn,
>
> We thank the reviewer for their thorough and constructive review, and for the careful assessment of both the strengths and limitations of our work. We appreciate the positive evaluation of the proposed approach. We are also grateful for the detailed feedback on the physical interpretation, clarity of presentation, and experimental validation. Below, we detail the revisions and additions made to address all raised concerns:
>
> ### Requested changes
>
> > 1. Discussion on how $\mathbf{M}_m$ and $\mathbf{V}'_m$ are represented as inputs to the hypernetwork $g$, and $\mathbf{r}_n$, as input to $f$.
>
> Thank you for pointing this out. We have now added a detailed description of the inputs to the hypernetwork for each experiment.
>
>
> > 2. Explicit mention of the consideration of static field representations
>
> The review is correct in pointing this out. We now explicitly mention on page 3 in the beginning of the Method section that we consider static configurations of sources, i.e. configurations where the locations of the source points do not change in time, but where the magnetization of the source points can change in time.
>
> > 3. The Ampère–Maxwell law needs to be more clearly presented.
>
> We agree that in a static setting both Gauss's law for magnetism and Ampère–Maxwell law for magnetism needs to be satisfied. We have now included the latter in the method section and specifically say that Eq. (1) is a consequence of the laws of magnetism.
>
>
> > 4. Where we get divergence-free fields from the constructions suggested in this work.
>
> That is correct. We have changed the respective bullet point from divergence-free fields to conservative fields, which is consistent with the rest of the manuscript.
>
> > 5. Some non-dimensionalization has been applied such that the permeability of free space is 1.
>
> We thank the reviewer for raising this point. No non-dimensionalization has been applied in Eq. (1): the magnetic field **H** and the magnetization **M** have the same physical units, so no factor of the permeability of free space is missing. The magnetic flux density **B**, which involves the permeability of the free space, is not used in this work.
>
> > 6. On the notation for the Euclidean norm
>
> We thank the reviewer for spotting this inconsistency in notation. We have changed the notation to that used in Eq. (12), i.e. $\parallel \cdot \parallel$, throughout the manuscript as well as defined it to be the Euclidian norm the first time the notation is used. This also removes any confusion with the use of $\vert \cdot \vert$ for the absolute value which also appears throughout the manuscript.
>
> > 7. The notation $\delta_{ij}$ is never defined.
>
> We now clearly state in the manuscript that  $\delta_{ij}$ is the Kronecker delta, which is equal to 1 if the indices are equal and 0 otherwise.
>
> > 8. A better unification and discussion of the Fourier expansion of the scalar potential.
>
> Thank you for making this apparent to us. We have harmonized the formulations of Eq. (10) and Eq. (13), which improved our descriptions of the Fourier network.
>
> > 9. Improving clarity in the discussion of $\mathbf{k}_l$ at the end of Section 3.3.
>
> We agree with the reviewer. We have added a comment that clarifies our choice for the wavenumbers. We are using the Fourier network exclusively in Section 5.2.1, where we explicitly state how $\mathbf{k}$ is instantiated.
>
> > 10. Neural operators and their relationship to this work should be discussed, at least briefly, in the Related Work section.
>
> We agree including a discussion on the similarities and differences here would benefit the manuscript. We have now added a section in Related Work to clarify this. Moreover, we have now added the Fourier neural operator framework as a baseline to predict the scalar potential from prism-shaped sources and compared its performance to our method in the Appendix.

---

> ### Author Response · Authors · 2025-12-14
> **Response to Reviewer 2oVn (2/4)**
>
> > 11. Justification and references on the relative error measure.
>
> As magnetic fields are relative, i.e. their strength depends on the oberserver's frame of reference, and therefore span a wide range of values, it is common to use the relative error for such problems. See Eq. 57 in *Hongwei Cheng, Leslie Greengard, and Vladimir Rokhlin. A fast adaptive multipole algorithm in three dimensions. Journal of Computational Physics, 155(2):468–498, 1999.*
>
> The choice of using the median relative error on the other hand was chosen deliberately for our work as we evaluate the point-wise error opposed to other works where the error is taken relative to the average norm across the domain. As can be seen in Fig. 9 in the Appendix, our way of measuring produces (very) large relative errors ($\gg100\%$) where the ground-truth potential values are close to 0, which results in a skewed mean error. The median error gives here a more representative measure of performance. For visualization purposes, we have given the point-wise error distributions in Fig. 6 (bottom left), Fig. 8, and Fig. 9.
>
> Additionally, we have now extended Table 3 with the total MAE in line with the way of measuring error in *R. Bjørk, E. B. Poulsen, K. K. Nielsen, et al. MagTense: A micromagnetic framework using the analytical demagnetization tensor. Journal of Magnetism and Magnetic Materials, 535:168057, October 2021.*
>
> > 12. Clarification of the training setup of Section 5.1.
>
> Thank you for pointing this out. We have clarified the specific training setup of Section 5.1 now in the manuscript. A specific setup similar to FC+ILR is used here for a single 2D potential / field produced by three magnetic sources, which is shown in Fig. 2a.
>
> > 13. Are curves in Figure 3a the validation errors as training proceeds (i.e. validation curves)?
>
> That is correct and we are indeed presenting validation curves here. This is corrected and clarified in the manuscript now.
>
> > 14. Is the "Linear model" the FC + ILR setup, but where the networks for $f$ and $g$ are restricted to linear transformations, as opposed to deep networks with non-linear activations?
>
> The "Linear model" differs from the FC + ILR setup in being restricted to linear transformations, as the reviewer correctly says, and, importantly, in being restricted to using single layers. We have added more information to Section 5.2.3 to make the setup of the "Linear model" more clear.
>
> > 15. Why is the height of the prism ($s_z$) relevant?
>
> In general, magnetostatics and the underlying phenoma are defined in 3D. In order to get a proper 2D potential / 2D field, i.e. $\phi_z=0, H_z = 0$ from a uniformly magnetised prism (defined naturally in 3D with MagTense), we need to set $M_z = 0$ for all sources and evaluate the potential / field in the midplane of the prism. To further ensure translational invariance in $z$-direction, i.e. $\phi_z=0,~H_z = 0$ away from the midplane, we have define an infinitely long prism by making $s_z \gg s_x, s_y$ in practice. This ensures also that the resulting potential / field is equivalent with a potential / field generated by a proper 2D rectangular source with side lengths $s_x$ and $s_y$ without having a $z$-component. However, as this is only relevant for the practival implementation in MagTense, mentioning this information in the beginning of Section 6  might be confusing and we refer the reader to the published code along with this work.
>
> > 16. The notation of $N_{tile}$ and $N^{(m)}_{tile}$ are introduced without explanation on Page 11. This makes interpreting and understanding the results in this section difficult.
>
> We have simplified the notation to $N_m$ and explicitly introduce the notation with "the number of evaluation points inside a source $m$".
>
> > 17. The training setups for Overlapping Sources and Quadtrees are never described.
>
> Detailed configurations of the training setup can be found in Table 4 in the Appendix. We have now also explicitly mentioned the training setups in the beginning of Sec. 6.
>
> > 18. Paragraph on Section 6.2 about failures associated with $M=250$. Could the model be trained on prisms in this size range to overcome this issue?
>
> We have reformulated the specific paragraph to emphasize what our currently trained model is capable of and what its limitations are. Increasing the size range will lead to similar issues discussed in the following point.

---

> ### Author Response · Authors · 2025-12-14
> **Response to Reviewer 2oVn (3/4)**
>
> > 19. Claim about dynamic range limitations for small prisms.
>
> We think that the experiments shown in Fig. 8 in the Appendix arguably support our claim. This figure is also referenced in the respective paragraph. Our training sizes range from 0.05 to 0.5, but nevertheless we see a drastic increase in the relative error for single-source validation samples in Fig. 8a, when the magnetic source becomes small ($s_x<0.3$) while still being well inside the training range. We further can see in Fig. 8b that samples with small sources ($s_x < 0.2$) perform worse across the whole range of the domain, also inside the source. For our training data, the strength of the 2D potential stretches over two orders of magnitude, which can be seen from Eq. 14 and the linear dependence on the magnetic moment $ {\mathbf{m}}_ m = 4 \cdot s_{mx}^2 \mathbf{M}_m$ (here for 2D prisms) with $0.05^2 \ll 0.5^2$.
> Across all our trained models we can see a similar behaviour, which let us assume that this is not easily remedied with a larger model or extended range of training data. In future work, one idea is to incorporate the known $1/\lVert\mathbf{r}\lVert$ scaling of the 2D far-field potential into the neural network architecture. This is discussed in the manuscript now.
>
>
> > 20. Experiments on the generalization across source geometries (cylinders to prisms, for example) without retraining, and on mixing different kinds of source geometries during training.
>
> The magnetic field generated by different source geometries is different, because the field near the object depends crucially on the geometrical shape of the object. Therefore, if the source geometry is different, the network needs to be retrained. The far field will approach the field of a dipole, so it is possible to envision a network that could be trained on different source geometries, but that it beyond the scope of this work. We have now added the discussion, including references, in the manuscript.
>
> > 21. In Appendix C.2, the importance of training on irregularly sampled points is stated, but in the experiments, it appears that all training data was sampled on a uniform grid. If this is not the case it should be made clear. Otherwise, the statements of this appendix need to be amended.
>
> For the experiments in Sec. 5.2 and Sec. 6, we have that the training data "is evaluated at $32^2$ uniformly sampled potential points", which should denote that we sample the evaluation points from a uniform distribution with the domain as intervals opposed to the regular grid mentioned in the Appendix. In Sec. 5.1, we indeed use a "regular $100^2$ grid", but this does not conflict with the statetment in Appendix C.2 for the Fourier model as only the FC+ILR model is used here. We have changed "uniformly" to "randomly across the domain" to make this more clear in the description of the training data.

---

> ### Author Response · Authors · 2025-12-14
> **Response to Reviewer 2oVn (4/4)**
>
> ### Clarifications and Minor Comments
>
> > 1. Motivation for using the Huber loss.
>
> We use the Huber loss instead of MSE because the target fields and potentials span a very large dynamic range across space, due to strong distance-dependent scaling and near-field / far-field effects. With MSE, training becomes dominated by a small number of large-magnitude errors close to sources, which leads to unstable optimisation and poorer performance away from the sources. The Huber loss mitigates this by behaving quadratically for small residuals while limiting the influence of large errors, providing a more robust objective. We have extended this motivation in the revised manuscript.
>
> > 2. When using a Fourier representation for the electric potential, could you use derivatives of the Fourier representation to derive the field, rather than relying on numerical differentiation?
>
> Yes, as the analytical derivatives of the trigonometric basis functions are well known, this is definitely possible. However, in Sec. 5.1 we use a version of the FC+ILR model.
>
> > 3. Typos on "supoerposition", "bracktes", and "because his model".
>
> Thanks for pointing these out! We've now corrected them.
>
> > 4. Table 1 appears quite far from it's first refence on Page 6.
>
> We agree it is far from the reference on page 6, however we also reference Table 1 in the introduction which is why we place it here.
>
> > 5. Other parameters, beyond learning rate, for the Adam optimizer are not specified anywhere.
>
> In addition to the learning rate, all other Adam hyperparameters are kept at their default values. Specifically, we use the standard Adam configuration with $\beta_1 = 0.9$, $\beta_2 = 0.999$, and $\epsilon = 10^{-8}$. We have added this clarification to the manuscript.
>
> > 6. On the "ensemble of fully-connected networks":
>
> We have clarified that the ensembles are uniformly averaged now in the manuscript.
>
> > 7. The activation functions for the dense networks are never stated anywhere.
>
> Throughout the experiments, we use Gaussian error linear unit (GELU) activation function. We have added this to Tabel 4 in the Appendix.
>
> > 8. Equation 13 has extra white space after the $\Sigma$s that should be fixed.
>
> Thanks, we've corrected the formatting here now.
>
> > 9. It would be helpful to know how the plots for the field and potential are created. That is, what kind of interpolation (or other approach) is used between the evaluation points?
>
> For the plotting of Figs. 2 and 4, we have used `matplotlib`'s `contour()` function with default arguments for the potential and `streamplot(..., density=1.5, linewidth=1, arrowsize=1.5, arrowstyle="->")` for the field.
>
> For Figs. 6 and 8, we have used `matplotlib`'s `contour(..., levels=250, norm=matplotlib.colors.Normalize(vmin=`$-|\phi_{\text{max}}|$`, vmax=`$|\phi_{\text{max}}|$`))` function with default arguments for the normalised potential and `streamplot(..., density=1.5, linewidth=0.5, arrowsize=1.5, arrowstyle="->")` for the field.
>
> This information is accessible via the Jupyter notebooks in the published code.
>
> > 10. Are the "10 runs" referred to in Section 6.1 ten separate training runs?
>
> Here, we actually want to refer to 10 different multi-source validation samples, which we now have made more clear in the manuscript.
>
> > 11. In Appendix B.1, the notation $\mathbf{m}_m$ clashes with $\phi(\mathbf{r}, \mathbf{M}_m, \mathbf{V}'_m)$.
>
> We have clarified that $\mathbf{m}_m$ is the magnetic moment of source $m$ (opposed to its magnetisation $\mathbf{M}_m$). Further, we give its definition for a spherical source that uniformly magnetized: $\mathbf{m}_m = \frac{4}{3} \pi d_m^3 \mathbf{M}_m$.
>
> > 12. Table 4 does not do a good job specifying what "Width" and "Depth" refer to. I think they are for $f$? Moreover, "Hnet" is not used anywhere in the body. Is this $g$?
>
> We thank the reviewer for spotting this. We have labeled the "Width" and "Depth" with the respective network names from Fig. 1.

---

### Review · Reviewer_oYMD · 2025-12-08

**Summary Of Contributions:**

The authors present a generative model designed to infer physical fields, specifically magnetic scalar potentials, from collections of physical sources. The core contribution is an architecture that enforces the physical principle of superposition through a hypernetwork design. A hypernetwork takes source features such as geometry and magnetization and outputs weights for a target network, which may be either a Fourier basis or a fully connected network. The weights from multiple sources are summed before the target network evaluates the field, allowing the model to scale linearly as O(M+N) in the number of sources and evaluation points. This effectively amortizes the cost of computing interaction matrices. The authors model the scalar potential φ and derive the magnetic field H via automatic differentiation to ensure the field is conservative. The method is compared against the Fast Multipole Method, and the authors demonstrate that their FC+ILR model generalizes to arbitrary numbers of sources despite being trained on single-source data.

**Audience:**

Yes

**Audience Explanation:**

The specific technique of using hypernetworks to output linear layer weights that enforce an additive inductive bias is a generalizable architectural pattern that could be relevant to researchers working on other linear physical systems such as acoustics and electrostatics.

**Broader Impact Concerns:**

I don’t have any concerns.

**Claims And Evidence:**

Yes

**Claims Explanation:**

The authors provide clear empirical evidence for their scaling claims and generalization capabilities. Figures 3b and 4 demonstrate that the model, trained on single sources, successfully predicts fields for multiple sources, validating the central claim that summing hypernetwork weights preserves physical superposition. Figure 5a provides runtime comparisons confirming the O(M+N) behavior compared to the classical O(M×N) scaling of direct summation, showing competitive crossover points with FMM. The authors are transparent about the method’s accuracy limitations, explicitly analyzing high error rates exceeding 10% for small sources in Section 6.2 and Appendix D, attributing this to dynamic range issues.

**Requested Changes:**

I recommend several changes to strengthen the submission, none of which are strictly critical for correctness but would significantly improve the context and quality of the paper. I don’t think any change is strictly necessary.

1) I strongly recommend a comparison to neural operators. The paper compares the proposed method primarily to FMM, a classical numerical method. I recommend adding other deep learning approaches for solving PDEs and fields, such as DeepONet and Fourier Neural Operators. These architectures also aim to learn resolution-invariant operators. The authors should discuss how their hypernetwork plus summation approach differs from these methods. If feasible, a small empirical comparison on the single-source task would be valuable; if not, a qualitative comparison in the Related Work section is necessary to contextualize the contribution within modern machine learning.

2) the paper notes that the model struggles with small sources due to the large dynamic range of the potential. The authors should discuss whether they considered training on the log-potential or using a specialized normalization scheme to compress this dynamic range. A brief explanation of why this approach was not pursued, or whether it was tried and failed, would be helpful for practitioners facing similar multiscale issues.

3) Ezperiments are conducted in 2D, and while the mathematics generalizes to 3D, the computational cost of the grid grows cubically. The authors should expand the discussion in the Limitations section regarding the memory bottleneck of the proposed FC+ILR method in 3D, addressing whether the hypernetwork size would need to grow significantly to capture 3D geometries such as tetrahedra or meshes.

4) In Eq. 5, the notation for the summation inside the function f should be verified and made explicit, showing clearly that the parameters of f are the sum of the outputs of g.

---

> ### Author Response · Authors · 2025-12-14
> **Response to Reviewer oYMD**
>
> Dear Reviewer oYMD,
>
> We thank the reviewer for their constructive and positive assessment of our work. We appreciate the recognition of the core contributions and scaling properties of the proposed approach, as well as the helpful suggestions for improving context and clarity. Below, we describe how we have addressed each requested change:
>
> > 1. Comparison to neural operators. How the hypernetwork plus summation approach differs from these methods.
>
> We agree that including a discussion on the similarities and differences between neural operators and our method will benefit the manuscript. We have now added a separate paragraph in the Related Work Section to specify the architectural differences to our method. Moreover, we have now added the Fourier neural operator framework as a baseline to predict the scalar potential from prism-shaped sources and compared its performance to our method in the Appendix.
>
> > 2. The authors should discuss whether they considered training on the log-potential or using a specialized normalization scheme to compress this dynamic range. A brief explanation of why this approach was not pursued, or whether it was tried and failed, would be helpful for practitioners facing similar multiscale issues.
>
> We are grateful for this comment as this exactly captures some parts of our internal discussion, which we have now mentioned explicitly in the Appendix C.3 to guide practitioners:
> - When modelling the log-potential, we get issues on how to differentiate between positive and negative values.
> - We have tried to train "correction" models for prism-shaped sources in Sec. 6 to predict the difference to a dipolar field produced by a circular source with the same magnetic moments (from FMM2D) following the structure of residual networks in order to remove the necessity to learn the $1/\Vert\mathbf{r}\Vert^2$ scaling of the 2D field.
> - Also, we have overpopulated the training data with small sources without success.
> - What has slightly remedied this problem was to multiply the values for the magnetisation drawn from a normal distribution by a factor of 10. As magnetic fields are relative, this factor can easily be removed by dividing model predictions by the same factor. This partly compensates for small-sized magnets and also points towards that our method has a preferred field range, where the chosen hyperparameter lead to a better model.
>
>
> > 3. Extension to 3D and scalability.
>
> That is a very important remark and we have extended to discussion in the Limitations section to shed more light on this matter. From an implementation perspective only the input layers of $\mathbf{\psi}$ and $g$ inevitably have to grow in size for the FC+ILR to incorporate additional geometric information and the magnetisation in 3D. From a learning perspective, we also expect the output layer size $L$ and hence the number of basis functions to increase, as does in a FOURIER setup when going from 2D to 3D, in order to properly model 3D potentials / fields. Additionally, intermediate layer szies potententially have to grow. However, without proper experiments in 3D, which is beyond the scope of this work, it is difficult to tell to which extend the hypernetwork size has to grow in 3D to have comparable performance to the presented results in 2D.
>
> > 4. Notation for the summation inside the function f in Eq. 5.
>
> Thank you for pointing this out. In Eq. 5, we have now explicitly specified function $f$ to make it more clear that the parameters are derived from $\mathbf{a}$, which is the sum of the outputs of $g$, and do not depend on the query locations $r_n$.

---

> > ### Comment · Reviewer_oYMD · 2026-01-07
> >
> > Thank you for your thorough revisions. I have no further requests and maintain my assessment that the paper's claims are well-supported.

---

### Review · Reviewer_6Gx1 · 2025-12-22

**Summary Of Contributions:**

This manuscript introduces a hypernetwork-based generative model for source-to-field inference in physical system such as magnetostatics and gravitation. It amortizes the evaluation of classical fields form complex source geometries and achieves O(M+N) scaling rather than traditional O(M*N). The paper propose two main model variants: Fourier hypernetworks and FC+ILR, and benchmark them against classical and machine learning baselines. The proposed method claims to maintain the principle of superposition and provide continuous field approximations across arbitrary source configurations.

**Audience:**

Yes

**Audience Explanation:**

The proposed use of hypernetworks to achieve O(M + N) inference for physical fields is novel and practically relevant for surrogate modeling. The idea of treating fields as continuous generative functions are of interest to a broad segment of the TMLR audience.

**Broader Impact Concerns:**

No.

**Claims And Evidence:**

No

**Claims Explanation:**

Supported Claims：
Claim 1: The model achieves O(M + N) scaling. Training cost and model size increase with Fourier mode count or network width, which is not fully addressed.
Claim 2: The model respects the superposition principle. Additive architecture explicitly aggregates source contributions.

Weakly or Unsubstantiated Claims：
Claim 3: The model generalizes to arbitrary source geometries and configurations. But the training is limited to simple 2D shapes. There are no experiments in 3D.
Claim 4: Accuracy is competitive with Fast Multipole Methods (FMM). FC+ILR outperforms FMM near source centers (Fig. 5b) but underperforms in quadtree structures (Table 3).

**Requested Changes:**

Major concern

1.	All experiments are restricted to 2D fields and simplified source geometries. Despite author claims that 3D extension is straightforward, this no cases to support it. It would be better to show at least one 3D case to make the method more practical.
2.	There is no online code to verify.
3.	Please address the limitation of the generalization across source scales which is shown in Table 3.
4.	The model is trained on paired analytical expression. While there are models that are lack of analytic solutions in physics. I recommend test model on data where only noise, partial, or simulated data are available.
5.	The formulas for f in Figure 1 include Eq(10) and Eq(11), there the expression in Figure 1 should be more generalized. Also the dimensions of tensor in Figure 1 should be more clear. Please use same symbols for the same quantities. And explain clearly which variable is a special form of which variable, such as Eq(10) and Eq(11).
6.	It is not clear how the networks are trained for all cases. Please clarify this for each cases, including the evaluation points for training and validation, number of pairs for training and validation, how to generate these pairs (for exact solution please show the ground truth).
7.	Is the same evaluation points used in training and validation? To verify the successful learning model need to use different evaluation points in training and validation.
8.	There is no visualizations of residuals, spatial error maps, or confidence intervals. Present worst-case examples and qualitative degradation in high-error regions.
9.	Have you ever tried non-uniform evaluation points, please try some to verify the successful training of continuous model or explain the reason why fails.
10.	When explain the computation expense, please also consider the computation of networks.
11.	Have you ever tried low resolution in training and high resolution in validation? This is a normal test for claimed continuous model learning.

Minor concern

1.	Section 3, Line3: “impute” -> “compute”
2.	Eq (5)(10)(13) are described too informally. Include full derivations in appendix.
3.	Figure 1 is central to understanding but contains cramped notation. Consider separating components (ψ, g, f) into distinct blocks with clearer labeling.
4.	Section 5.1: What is the g as you said “we train f while keeping g fixed”
5.	Explain the initialization of the networks when training.

---

> ### Author Response · Authors · 2026-01-05
> **Response to Reviewer 6Gx1 (1/2)**
>
> We thank the reviewer for their constructive feedback. We appreciate their acknowledgement of the core contributions and scaling behaviour of the proposed method, as well as their suggestions for improving clarity and context to support the claims of our submission. We address each requested change below.
>
> > 1. All experiments are restricted to 2D fields and simplified source geometries. Despite author claims that 3D extension is straightforward, this no cases to support it. It would be better to show at least one 3D case to make the method more practical.
>
> In the discussion section, we have now explicitly mentioned how our method (FC+ILR) can be extended to 3D in practice. Further, from a physical perspective, if the method works in 2D, it must work in 3D as well. However, from a learning perspective, we need to find the correct network sizes and optimiser settings, while possibly having to adjust other hyperparameters, e.g. $\gamma_{\mathbf{H}}$. Further, one would have to cope with the increased dataset size when keeping the resolution at 32, also for the $z$-component. Hence, we argue that the 3D extension is beyond the scope of this work.
>
> > 2. There is no online code to verify.
>
> We disagree with this statement, which must result from an oversight of the reviewer. We provide links to our code in the abstract and later at the beginning of Section 5.
>
> > 3. Please address the limitation of the generalisation across source scales, which is shown in Table 3.
>
> In Table 3, we show how our method generalises from training, where only single-source samples are available, to experiments with multi-source samples. Here, we only see a slight increase in the relative error $\epsilon_{\phi}$ from 2.21% ($M=1$) to 4.21% ($M=1000$). However, as you mention and as we discuss in Appendix D, we notice a substantial difference in $\epsilon_{\phi}$, when comparing small-sized sources ($s_x < 0.2$ and $\epsilon_{\phi} >10$%) with large sources ($s_x > 0.4$ and $\epsilon_{\phi} < 5$%). We have addressed this more in depth now in the last paragraph of Section 6.
>
> > 4. The model is trained on paired analytical expression. While there are models that are lack of analytic solutions in physics. I recommend test model on data where only noise, partial, or simulated data are available.
>
> As the aim of this paper is to approximate Eq. 1 with reduced computational complexity $\mathcal{O}(M+N)$, where we infer the correct magnetic field at arbitrary query locations for a given magnetic source configuration, we are comparing how well our method approximates the true physics. We agree that it would be interesting to use our model in downstream tasks, e.g. inverse design with noisy measurements, but this is beyond the scope of this work.
>
> > 5. The formulas for f in Figure 1 include Eq(10) and Eq(11), there the expression in Figure 1 should be more generalized. Also the dimensions of tensor in Figure 1 should be more clear. Please use same symbols for the same quantities. And explain clearly which variable is a special form of which variable, such as Eq(10) and Eq(11).
>
> We have aligned the respective equations to exactly match with the general expression of f in Fig. 1. We have also made more clear how a special form of a variable is connected to the general expression, e.g. $\mathbf{A}_{p}$ corresponds to $\mathbf{a}$ in Eq. 10. We also describe the order of the FOURIER network now with P, so that the dimensions of tensors of $\boldsymbol{\psi}: \mathbb{R}^d \rightarrow \mathbb{R}^L$ and $\mathbf{a} \in \mathbb{R}^L$ are consistent throughout the manuscript. We mention now the dimension of tensors whenever possible. Further, we have introduced $v$ as the dimension of the source features as input to $g$.
>
> > 6. It is not clear how the networks are trained for all cases. Please clarify this for each cases, including the evaluation points for training and validation, number of pairs for training and validation, how to generate these pairs (for exact solution please show the ground truth).
>
> We have extended the description of the training setup for all cases to clarify the experiments and to ensure reproducibility.
>
> > 7. Is the same evaluation points used in training and validation? To verify the successful learning model need to use different evaluation points in training and validation.
>
> We are aware of this fact, and we have used different evaluation points for training and validation throughout our work. We have rephrased the respective parts in the manuscript to make this very clear.

---

> ### Author Response · Authors · 2026-01-05
> **Response to Reviewer 6Gx1 (2/2)**
>
> > 8. There is no visualizations of residuals, spatial error maps, or confidence intervals. Present worst-case examples and qualitative degradation in high-error regions.
>
> We do not agree with this comment, as we specifically visualise residuals and spatial error maps of our method in, e.g. Fig. 6. Further, worst-case examples and qualitative degradation in high-error regions, which occur mostly in predictions of the scalar potential of small-sized sources, are presented in Fig. 9.
>
> > 9. Have you ever tried non-uniform evaluation points, please try some to verify the successful training of continuous model or explain the reason why fails.
>
> As our method is naturally a continuous model, without the need for discretisation, we choose uniform evaluation points merely for visualisation purposes. For example, the results in Table 3 hold irrespective of the choice of uniform or non-uniform evaluation points. We have made this clearer now when discussing the result of our model.
>
> > 10. When explain the computation expense, please also consider the computation of networks.
>
> That is indeed a very interesting point of discussion. When we present the runtime in Fig. 5, we hence compare our method (and its underlying computation of networks) to the benchmark methods, i.e. MagTense and FMM. Further, we show the advantages of parallelisation in neural network computation on GPUs with the solid lines in Fig. 5a.
>
> > 11. Have you ever tried low resolution in training and high resolution in validation? This is a normal test for claimed continuous model learning.
>
> As mentioned earlier, our architecture does not rely on discretisation, i.e. the input to $\boldsymbol{\psi}$ is a single query location $\mathbf{r}_n$ and the input to $g$ is the source features in a parametric description, which is independent of discretisation. We train on randomly sampled points across the 2D domain. Validation can then be done at arbitrary resolutions. We have clarified this now in the manuscript and explicitly pointed out how this compares to neural operator learning.
>
> **Minor concern**
> > 1. Section 3, Line3: “impute” -> “compute”
>
> Thank you for pointing this out. We have changed it to "query", as it captures the exact meaning slightly better than "compute".
>
> > 2. Eq (5)(10)(13) are described too informally. Include full derivations in appendix.
>
> We have harmonised the formulations of Eq. 10 and Eq. 13, while also introducing some additional descriptions to make these more formal. In Eq. 5, we have now explicitly stated how $f$ is exactly represented with our method. We think this can be understood without further derivations in the appendix.
>
> > 3. Figure 1 is central to understanding but contains cramped notation. Consider separating components (ψ, g, f) into distinct blocks with clearer labeling.
>
> As the components (ψ, g, f) are represented as different blocks in Fig. 1 and are labelled respectively, we extended the caption of Fig. 1 to make each part of this figure more clear.
>
> > 4. Section 5.1: What is the g as you said “we train f while keeping g fixed”
>
> We have rephrased that paragraph to be more specific about what $g$ is here. We train on a single multi-source sample where we learn the optimal basis weights $a$ instead of inferring them from the source features.
>
> > 5. Explain the initialization of the networks when training.
>
> We have now stated explicitly in Table 4, how the networks are initialised. Moreover, this information can be found in the code published alongside the manuscript.

---

### Author Response · Authors · 2026-01-06

Hello,

We would like to thank the reviewers for their time and for the valuable feedback! Apologies for the delay in responding. We had set the visibility incorrectly on our replies, but hopefully the reviewers can see them now.

Thanks again and best wishes.

---

### Decision · Action_Editor_hyWw · 2026-01-21

**Recommendation:** Accept with minor revision

**Audience:**

Yes

**Audience Explanation:**

The paper addresses a relevant and non-trivial problem: the reduction of the complexity of physical field inference from quadratic to linear is a significant practical contribution. The use of hypernetworks to produce implicit representations of fields generalizes across source configurations and allows arbitrary evaluation points. The topic sits at an intersection of machine learning and computational physics, which is of interest to communities in scalable simulation and physics-informed learning. So from a scientific and ML standpoint, the problem is interesting and potentially impactful, provided the empirical evidence and comparisons are solid. All reviewers strongly agree on this point.

**Claims And Evidence:**

Yes

**Claims Explanation:**

All reviewers agree that the paper meets the bar in terms of evidence, with two “Accept” recommendations and one “Leaning Accept” contingent on minor-to-moderate revisions. (“All my concerns are addressed.”, “technically solid and conceptually well-motivated”).

The revision is viewed as solid (including the added FNO baseline and expanded related work), and the reviewers see the paper’s transparency about limitations (dynamic range issues and negative results such as failed log-potential variants) as strength. Remaining evidence gaps primarily concern scope rather than correctness: experiments only cover uniform source geometries and 2D, so strong statements about generalization across geometries without retraining and practical 3D applicability could be supported with additional experiments; I still see the paper as strong enough for TMLR without these additions. The manuscript would also benefit from earlier, clearer articulation of magnetostatic/quasi-static assumptions and minor clarifications of terminology and figures, which is why I give the score "Accept with minor revision".

---

> ### Author Response · Authors · 2026-02-04
>
> We thank the Action Editor for the positive assessment and for the clear summary motivating the "Accept with minor revision" decision. We are pleased that the reviewers found the paper technically solid, well-motivated, and adequately supported by evidence, and that the transparency regarding limitations and negative results was viewed as a strength.
>
> We have made a small number of final clarifications to further improve clarity in terminology and figures, in line with the Action Editor's suggestions. In particular:
> - We have updated the caption of Figure 5 to clarify that $\epsilon_{\phi_n}$ refers to error values at individual evaluation points, rather than errors averaged over a sample or spatial domain.
> - We have clarified that the axes in Figures 4 and 6 are dimensionless and now explicitly state in the labels that spatial quantities are expressed in units of the source radius.
>
> In addition, we have ensured that the magnetostatic / quasi-static assumptions are stated clearly and early in the manuscript.